# ExpAlign: Expectation-Guided Vision–Language Alignment for Open-Vocabulary Grounding

Junyi Hu [1]   Tian Bai [2 3]   Fengyi Wu [2]   Wenyan Li [4]   Zhenming Peng [2]   Yi Zhang [1]

## Abstract

Open-vocabulary grounding requires accurate vision-language alignment under weak supervision, yet existing methods either rely on global sentence embeddings that lack fine-grained expressiveness or introduce token-level alignment with explicit supervision or heavy cross-attention designs. We propose **ExpAlign**, a theoretically grounded vision-language alignment framework built on a principled multiple instance learning formulation. ExpAlign introduces an Expectation Alignment Head that performs attention-based soft MIL pooling over token-region similarities, enabling implicit token and instance selection without additional annotations. To further stabilize alignment learning, we develop an energy-based multi-scale consistency regularization scheme, including a Top-K multi-positive contrastive objective and a Geometry-Aware Consistency Objective derived from a Lagrangian-constrained free-energy minimization. Extensive experiments show that ExpAlign consistently improves open-vocabulary detection and zero-shot instance segmentation, particularly on long-tail categories. Most notably, it achieves 36.2 $AP_r$ on the LVIS minival split, outperforming other state-of-the-art methods at comparable model scale, while remaining lightweight and inference-efficient. Code is aveliable at https://github.com/inlmouse/ExpAlign.

[1]Department of Automation, Tsinghua University, China [2]School of Information and Communication Engineering, University of Electronic Science and Technology of China, China [3]Linsulabs, China [4]University of Copenhagen, Denmark. Correspondence to: Yi Zhang <zhyi@mail.tsinghua.edu.cn>.

*Proceedings of the $43^{rd}$ International Conference on Machine Learning*, Seoul, South Korea. PMLR 306, 2026. Copyright 2026 by the author(s).

## 1. Introduction

Large vision-language models (VLMs) enable powerful zero-shot transfer by aligning images and texts in a shared embedding space (Radford et al., 2021; Li et al., 2023; Jia et al., 2021). Despite significant progress, precise spatial grounding, which involves localizing free-form textual concepts within images, remains a key challenge in dense prediction tasks such as open-vocabulary detection and segmentation (Kamath et al., 2021; Cai et al., 2022). Recent open-vocabulary methods (Liu et al., 2024; Cheng et al., 2024; Wang et al., 2025; Fu et al., 2025) alleviate vocabulary constraints but often struggle with complex linguistic phenomena, including negation, relations, and compositional descriptions, when fine-grained localization is required.

Recent theoretical analysis reveals an inherent limitation of CLIP-style joint embeddings: collapsing a prompt into a single global representation cannot simultaneously encode attribute binding, spatial relations, and negation under cosine similarity (Kang et al., 2025b). This geometric bottleneck motivates *token-level* vision-language alignment, where informative tokens are selectively emphasized rather than uniformly aggregated. However, incorporating token-level reasoning into dense grounding remains nontrivial due to weak supervision and optimization instability.

We propose **ExpAlign**, an expectation-guided vision-language alignment framework for open-vocabulary grounding. At its core is the **Expectation Alignment Head (EAH)**, which produces prompt-conditioned spatial alignment maps by aggregating token-wise similarities through a soft expectation mechanism. By treating spatial locations as latent instances and textual tokens as competing hypotheses, EAH performs implicit token selection *without instance-level annotations*, admitting a natural interpretation as attention-based soft pooling in multiple instance learning (MIL) (Ilse et al., 2018).

To further improve discriminability and spatial coherence, we introduce two auxiliary objectives. A **multi-positive InfoNCE loss** enforces prompt-level semantic separation under weak supervision, while a **Geometry-Aware Consistency Objective (GACO)** regularizes alignment maps by emphasizing relatively consistent regions within each

ground-truth mask. Together, they stabilize optimization and support both positive and negative prompts.

Experiments on LVIS (Gupta et al., 2019), ODinW (Li et al., 2022), and RefCOCO/+/g (Yu et al., 2016) show that ExpAlign delivers strong open-vocabulary detection and segmentation performance under similar pre-training scale and model capacity to recent baselines. It achieves competitive or superior results on LVIS rare categories and ODinW subsets, while on RefCOCO/+/g it outperforms detection-focused methods such as YOLOE but trails specialized grounding models like Grounding DINO-T due to CLIP's limited spatial reasoning.

## 2. Related Work

**Sentence-level vision-language Alignment.** vision-language pretraining methods such as CLIP (Radford et al., 2021) and BLIP (Li et al., 2023) align whole images with global text embeddings using contrastive objectives. These sentence-level alignment techniques have enabled strong zero-shot transfer for retrieval and classification, and have been adapted to open-vocabulary detection by using prompt embeddings as classifier proxies (Zhou et al., 2022). However, collapsing a prompt into a single vector can lose internal structure and limits fine-grained localization, motivating methods that exploit richer textual and visual interactions.

**Fine-grained and Multi-level Alignment.** To capture detailed cross-modal semantics, recent literature extensively explores vision–language alignment at the token, phrase, or multi-level granularity. For instance, GLIP and its variants (Zhang et al., 2022) unify object localization and phrase grounding via region–word contrastive learning, while X-VLM (Zeng et al., 2021) performs multi-grained pretraining by aligning text with visual concepts at the patch and concept levels. Similarly, studies in temporal grounding emphasize the necessity of exploiting fine-grained, word-level signals rather than treating all tokens uniformly under cross-modal attention (Kang et al., 2025a). To improve efficiency, late interaction paradigms like FILIP (Yao et al., 2021) bypass heavy cross-attention by computing token-wise similarities between text and image patches via max pooling, which MulCLIP (Truong et al., 2025) further extends into a multi-level alignment framework for long-context models. While these approaches underscore the importance of sub-global interactions, they typically rely on explicit cross-attention architectures, dense phrase annotations, or heuristic pooling operations. In contrast, ExpAlign leverages expectation-based aggregation to softly pool token similarities into spatial alignment maps. This approach preserves token discriminability under weak supervision, presenting a fundamentally distinct design and objective.

**Alignment Regularization and Objectives.** Contrastive learning remains a core tool for vision–language alignment, with InfoNCE-style objectives widely adopted for separating positive and negative pairs in multimodal settings (Oord et al., 2018; Li et al., 2023). Region-level contrastive losses have been proposed to improve localization quality (Zhang et al., 2022; Zhong et al., 2022), and dense alignment objectives have been incorporated into grounding frameworks to better capture spatial semantics. Our multi-positive InfoNCE adapts these ideas to multi-prompt supervision, focusing on the most informative regions. In addition, geometry-aware regularization has been explored in segmentation and structured prediction (Liang et al., 2021), but existing approaches typically rely on absolute geometric cues. In contrast, our geometry-aware consistency objective operates on relative instance statistics, encouraging coherent alignment without rigid spatial targets.

**RL-Inspired Regularization in VLMs** Several works leverage reinforcement learning (RL)-inspired techniques or loss functions as regularizers to improve robustness and generalization in vision-language models. For instance, Group Relative Policy Optimization (GRPO) (Shao et al., 2024) introduces an efficient PPO variant that computes advantages via group-relative ranking, inspiring advantage-weighted alignment mechanisms. VARP (Singh et al., 2025) uses agent-regularized preferences in RL from VLM feedback to better align rewards and mitigate inaccuracies. PRLL (Zheng et al.) applies LLM-assisted policy regularization for reward shaping, enabling adaptation in unfamiliar environments. In VLM fine-tuning, RL4VLM (Zhai et al., 2024) directly optimizes VLMs with regularization for decision-making, while VLM-RL (Huang et al., 2025) incorporates contrastive language goals as regularized rewards in autonomous driving. These works illustrate the increasing adoption of RL-based regularization to enforce consistency and reduce overfitting in multimodal settings.

ExpAlign advances vision–language grounding by combining soft token-level aggregation with principled regularization, balancing expressiveness and optimization stability. It situates itself between sentence-level and structured alignment methods by enabling fine-grained, supervision-efficient alignment without reliance on heavy cross-attention or explicit token annotations.

## 3. Method

### 3.1. Overview

We study open-vocabulary grounding, where a model aligns visual regions with flexible language prompts and produces region-level predictions for detection or segmentation. Given an image $I$ and a set of textual prompts $\{T_k\}_{k=1}^K$, our goal is to compute prompt-conditioned spatial alignment

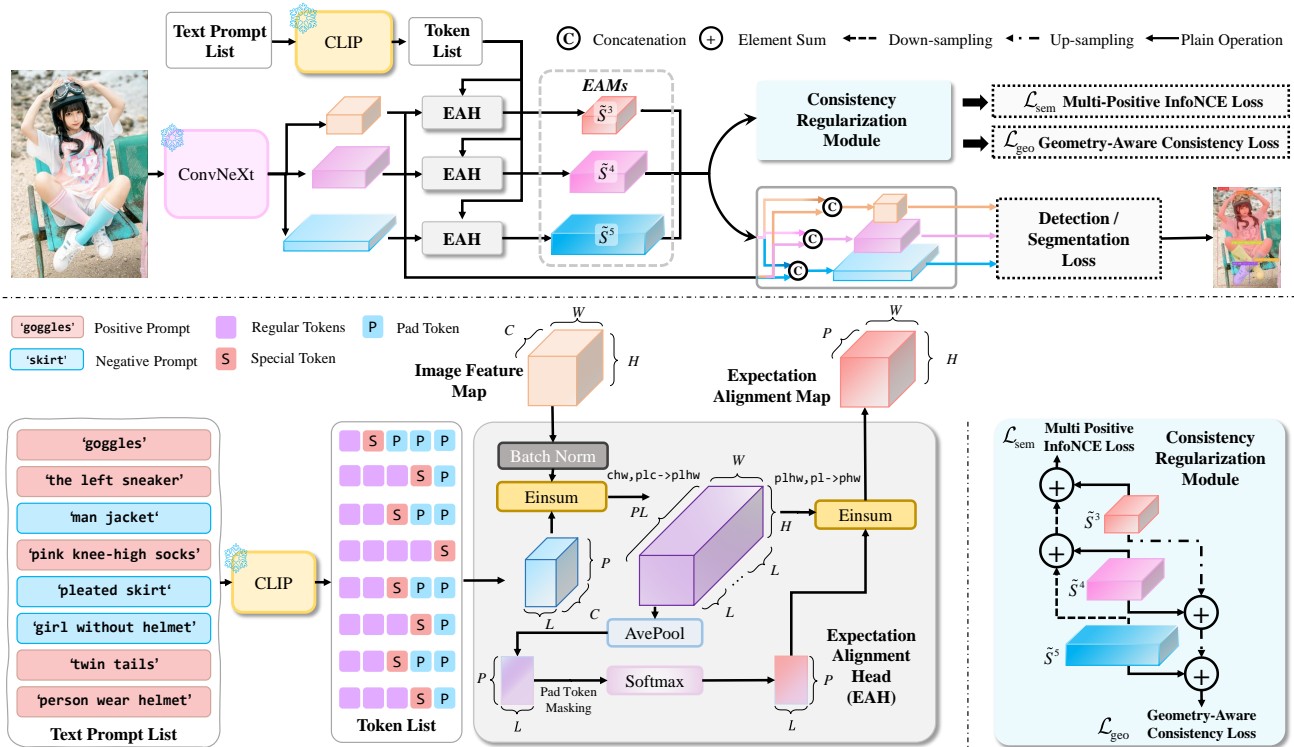

*Figure 1.* Overview of the proposed ExpAlign framework. **Top**: the overall pipeline, where prompt-conditioned Expectation Alignment Maps (EAMs) are computed at multiple feature scales and injected into visual features for open-vocabulary grounding and segmentation. **Bottom-left**: the Expectation Alignment Head, which aggregates token-level vision-language similarities into prompt-specific spatial alignment maps via expectation-based token weighting. **Bottom-right**: the Consistency Regularization Module, which applies semantic and geometric constraints to regularize the alignment maps. Best viewed in color.

maps that support robust localization under ambiguous and weak supervision.

We propose **ExpAlign**, an expectation-guided vision-language alignment framework. As illustrated in Fig. 1, ExpAlign consists of three key components: (i) an *Expectation Alignment Head (EAH)* that performs soft prompt–region alignment via token-level expectation, producing an *Expectation Alignment Map (EAM)*; (ii) a *Consistency Regularization Module* that enforces cross-scale coherence of alignment maps; and (iii) auxiliary objectives that impose semantic and geometric constraints on the alignment distribution.

This design enables implicit instance selection over both prompt tokens and spatial locations, while remaining fully differentiable and compatible with standard detection and segmentation heads.

### 3.2. Expectation Alignment Head

For scale $s \in \{3, 4, 5\}$ the backbone produces a feature map $\mathbf{F}_b^s \in \mathbb{R}^{C \times H_s \times W_s}$, and each prompt $p \in \{1, \cdots, P\}$ (in image $b \in \{1, \cdots, B\}$) is represented by $L$ token embeddings $\mathbf{T}_{b,p} \in \mathbb{R}^{L \times C}$.

We compute the token-wise similarity at every spatial location $(x, y)$:

$$S_{b,p}^s(x,y,l) = \langle \mathbf{F}_b^s(x,y), \mathbf{T}_{b,p}(l) \rangle, \ l = 1, \ldots, L. \quad (1)$$

To estimate the global relevance of each token, we aggregate spatial evidence by average pooling:

$$\bar{S}_{b,p}^s(l) = \frac{1}{H_s W_s} \sum_{x=1}^{H_s} \sum_{y=1}^{W_s} S_{b,p}^s(x,y,l). \quad (2)$$

We then form a token posterior distribution via a softmax over non-pad tokens:

$$\pi_{b,p}^s(l) = \frac{\exp\big(\bar{S}_{b,p}^s(l)/\tau_t\big)}{\sum_{l'} \exp\big(\bar{S}_{b,p}^s(l')/\tau_t\big)}, \quad (3)$$

where $\tau_t$ is a temperature parameter.

Finally, we compute the expectation alignment map (EAM) by marginalizing over tokens:

$$\tilde{S}_{b,p}^s(x,y) = \sum_{l=1}^{L} \pi_{b,p}^s(l)\, S_{b,p}^s(x,y,l). \quad (4)$$

The resulting map $\tilde{S}_{b,p}^s \in \mathbb{R}^{H_s \times W_s}$ is referred to as the

*Expectation Alignment Map (EAM)* at scale $s$. This formulation performs implicit token selection by assigning higher weights to globally informative tokens, while suppressing noisy or irrelevant ones, and yields a spatial alignment score suitable for downstream grounding.

### 3.3. Consistency Regularization Module

Given the expectation alignment maps (EAMs) produced at multiple feature scales, we impose consistency constraints to stabilize vision-language alignment across scales while respecting their distinct computational and geometric properties. Specifically, EAMs from different scales are used in a scale-aware manner: low-resolution maps are employed for efficient semantic alignment aggregation, whereas high-resolution maps are preserved for geometry-sensitive consistency regularization.

**Semantic Constraint.** To aggregate semantic evidence across scales while maintaining efficiency, we first unify multi-scale expectation alignment maps (EAMs) at the coarsest resolution. Specifically, EAMs from all feature levels are progressively downsampled to the spatial resolution of the smallest scale (e.g., P5) and summed to obtain a unified alignment map:

$$\tilde{S}_{b,p}^{\mathrm{dw}} = \Big(\mathrm{Down}\Big((\mathrm{Down}(\tilde{S}_{b,p}^3)+\tilde{S}_{b,p}^4)/2\Big) + \tilde{S}_{b,p}^5\Big)/2. \quad (5)$$

where $\mathrm{Down}(\cdot)$ denotes resolution-aligned downsampling.

For each image $b$ and prompt $p$, we select the top-1% highest responses from the unified map:

$$\mathcal{I}_{b,p} = \mathrm{TopK}\Big(\tilde{S}_{b,p}^{\mathrm{dw}}, \ H_3 W_3/100\Big). \quad (6)$$

We then define the pooled prompt-level logit as the average alignment score over these selected locations:

$$\ell_{b,p} = \frac{1}{|\mathcal{I}_{b,p}|} \sum_{i \in \mathcal{I}_{b,p}} \tilde{S}_{b,p}^{\mathrm{dw}}(i). \quad (7)$$

Let $\mathcal{P}_b$ denote the set of positive prompts for image $b$. The multi-positive InfoNCE objective with temperature $\tau$ is formulated as:

$$\mathcal{L}_{\mathrm{sem}} = -\frac{1}{B}\sum_{b=1}^{B}\sum_{p \in \mathcal{P}_b}\frac{1}{|\mathcal{P}_b|}\log\frac{\exp(\ell_{b,p}/\tau)}{\sum_{p'=1}^{P}\exp(\ell_{b,p'}/\tau)}. \quad (8)$$

**Geometry Constraint.** We introduce a *Geometry-Aware Consistency Objective (GACO)* to regularize the spatial structure of the energy field produced by the Consistency Regularization Module. Instead of enforcing absolute geometric targets, GACO shapes the energy landscape through relative, instance-wise consistency within the ground-truth region.

We construct a unified high-resolution Expectation Alignment Map by progressively aggregating EAMs from coarse to fine levels. Starting from the coarsest scale (P5), we apply a top-down fusion strategy:

$$\tilde{S}_{b,p}^{\mathrm{up}} = \big(\mathrm{Up}\big((\mathrm{Up}(S_{b,p}^5) + S_{b,p}^4)/2\big) + S_{b,p}^3\big)/2. \quad (9)$$

where $\mathrm{Up}(\cdot)$ denotes resolution-aligned upsampling. This aggregation preserves fine-grained geometry while incorporating multi-scale semantic evidence.

Given the aggregated alignment map, we define a normalized distribution over prompt–patch pairs by applying a softmax over all prompts and spatial locations:

$$\mathbb{P}_b(p,i) = \frac{\exp\big(\tilde{S}_{b,p}^{\mathrm{up}}(i)\big)}{\sum_{p'=1}^{P}\sum_{i'\in\Omega}\exp\big(\tilde{S}_{b,p'}^{\mathrm{up}}(i')\big)}, \quad (10)$$

where $i$ indexes spatial locations at the P3 resolution. This distribution assigns higher probability mass to patches that are more strongly aligned with a given prompt.

Let $M_{b,p}(i) \in \{0,1\}$ denote the binary ground-truth mask associated with prompt $p$, resized to the P3 resolution, and define the positive region $\mathcal{M}_{b,p} = \{\, i \mid M_{b,p}(i) = 1 \,\}$.

We introduce a bounded local alignment confidence

$$R_b(p,i) = \sigma\big(\tilde{S}_{b,p}^{\mathrm{up}}(i)\big), \quad (11)$$

and compute its mean and standard deviation over the positive region:

$$\mu_{b,p} = \frac{1}{|\mathcal{M}_{b,p}|}\sum_{i\in\mathcal{M}_{b,p}} R_b(p,i),$$
$$\sigma_{b,p} = \sqrt{\frac{1}{|\mathcal{M}_{b,p}|}\sum_{i\in\mathcal{M}_{b,p}}\big(R_b(p,i)-\mu_{b,p}\big)^2 + \varepsilon}. \quad (12)$$

Based on these statistics, we define a *relative consistency score*:

$$A_{b,p}(i) = \mathrm{clip}\bigg(\frac{R_b(p,i)-\mu_{b,p}}{\sigma_{b,p}}, \ -c, \ c\bigg), \quad (13)$$

which measures how well each patch agrees with the dominant spatial structure of its corresponding ground-truth instance. Importantly, this score depends only on intra-instance relative statistics and does not impose absolute alignment targets.

The final geometry-aware consistency loss is defined as

$$\mathcal{L}_{\mathrm{geo}} = -\frac{1}{\sum_b\sum_p |\mathcal{M}_{b,p}|}\sum_{b=1}^{B}\sum_{p=1}^{P}\sum_{i\in\mathcal{M}_{b,p}} A_{b,p}(i)\log\mathbb{P}_b(p,i). \quad (14)$$

This objective redistributes probability mass within each ground-truth region according to relative geometric consistency, encouraging spatially coherent alignment maps while remaining invariant to monotonic transformations of alignment scores. Rather than collapsing responses via pointwise regression, GACO sculpts the geometry of alignment maps and naturally supports implicit instance selection.

Notably, the variational framework (Appendix B) provides principled constraints that guide the concrete design of the geometry score. Specifically, it requires that any valid geometry score $A_{b,p}(i)$ must satisfy three properties: (i) depend only on intra-instance relative statistics, (ii) be bounded, and (iii) be locally Lipschitz. The clipped z-score formulation in Eq. 13 is the simplest, numerically stable, and differentiable realization that fulfills all three requirements. While the transition from the variational derivation to this concrete form is motivated rather than a strict algebraic derivation, the theoretical framework directly constrains and motivates each design choice, preventing simpler but less effective alternatives (e.g., global normalization or pointwise regression) that would violate these properties. The full variational derivation is provided in the Appendix.

### 3.4. Full Training Objective

The proposed consistency regularization serves as auxiliary supervision during training, complementing the standard detection or segmentation objective. Specifically, the semantic consistency loss $\mathcal{L}_{\text{sem}}$ promotes instance-level prompt–patch alignment, while the geometry-aware consistency loss $\mathcal{L}_{\text{geo}}$ encourages spatially coherent responses within each ground-truth region. Let $\mathcal{L}_{\text{det/seg}}$ denote the task-specific loss (including classification, regression, and mask prediction terms when applicable). The overall training objective is:

$$\mathcal{L} = \mathcal{L}_{\text{det/seg}} + \lambda_{\text{sem}}\,\mathcal{L}_{\text{sem}} + \lambda_{\text{geo}}\,\mathcal{L}_{\text{geo}}. \quad (15)$$

Both consistency losses are used only during training and are discarded at inference time, preserving standard prediction behavior while improving vision-language alignment and spatial consistency.

### 3.5. Connection to Multiple Instance Learning

Our Expectation Alignment Map (EAM) is mathematically equivalent to attention-based soft pooling in multiple instance learning (Ilse et al., 2018). Concretely, by treating each spatial location as an instance and each textual prompt as a bag, the EAM implements a soft-MIL pooling over token hypotheses, which grants permutation invariance and expressive pooling power. The multi-positive InfoNCE loss and the Geometry-Aware Consistency Objective (GACO) further enforce discriminative bag semantics and regularize instance relationships within positive regions. A formal proof is provided in Appendix A.

## 4. Experiment

### 4.1. Implementation Details

**Model.** ExpAlign is implemented as a lightweight vision-language alignment module that can be seamlessly integrated into standard multi-scale detection and segmentation architectures. Unless otherwise specified, we adopt a frozen DINOv3 (Siméoni et al., 2025) ConvNeXt-T image encoder as the visual backbone. Following the encoder, we employ the same YOLOv8 (Varghese & Sambath, 2024) FPN-style feature enhancement module to produce multi-scale feature maps. The detection and segmentation heads, along with their corresponding loss functions, strictly follow the standard YOLOv8 formulation without modification.

Text prompts are encoded using a frozen CLIP (Radford et al., 2021) ViT-L/14 text encoder, where we retain all token-level representations before the end-of-text (EOT) token, rather than collapsing the prompt into a single global embedding. To map textual tokens into the same feature space as visual representations, we append a lightweight Residual SwiGLU feed-forward network (SwiGLUFFN) (Shazeer, 2020) after the CLIP text encoder. The second linear layer of the SwiGLUFFN is initialized to zero, such that the module initially behaves as an identity mapping. This design stabilizes early training and ensures that token-level alignment is learned progressively without disrupting the pretrained CLIP geometry.

The Expectation Alignment Head (EAH) is attached to each feature level and computes prompt-conditioned alignment maps via token-wise similarity aggregation. The Consistency Regularization Module operates solely on the resulting alignment maps and introduces no additional learnable parameters.

**Data.** We adopt the same data protocol as Cheng et al. (2024) and train ExpAlign on a combination of detection and grounding datasets. Specifically, we use Objects365 (Shao et al., 2019) for large-scale object detection and GoldG (Kamath et al., 2021), which aggregates GQA (Hudson & Manning, 2019) and Flickr30k (Plummer et al., 2015), for vision-language grounding. To avoid data leakage, all images overlapping with COCO (Lin et al., 2014) are excluded from the training set. Since pixel-level annotations are not available for most training images, we generate pseudo instance masks for segmentation by applying the SAM-2.1 model (Ravi et al., 2024) to ground-truth bounding boxes from the detection and grounding datasets.

**Training.** All experiments employed the AdamW optimizer combined with a cosine learning rate scheduler across a two-stage training procedure. Both training and evaluation were carried out on a dedicated machine featuring eight NVIDIA RTX Pro 6000 GPUs, each with 96 GB of memory. Both the image encoder and the text encoder remain frozen

*Table 1.* **Zero-shot detection performance.** Metrics on LVIS val (Gupta et al., 2019) and minival (Kamath et al., 2021) are fixed AP (Achal Dave et al., 2022). All models use an input resolution of 640×640, except for those with Swin-Tiny as the backbone, which employ 800×1333. For training data, OG indicates Objects365 (Shao et al., 2019) and GoldG (Kamath et al., 2021). RefC indicates RefCOCO/g/+ (Yu et al., 2016).

| Method | Backbone | Pre-train Data | #Params | LVIS$^{minival}$ | | | | LVIS | | | | ODinW13 | ODinW35 |
| | | | | AP | AP$_r$ | AP$_c$ | AP$_f$ | AP | AP$_r$ | AP$_c$ | AP$_f$ | AP | AP |
|---|---|---|---|---|---|---|---|---|---|---|---|---|---|
| GLIP-T (Li et al., 2022) | Swin-T | OG | 232M | 24.9 | 17.7 | 19.5 | 31.0 | 16.5 | 7.5 | 11.6 | 26.1 | - | - |
| DetCLIP (Yao et al., 2022) | Swin-T | OG | - | 34.4 | 26.9 | 33.9 | 36.3 | - | - | - | - | - | - |
| GDINO-T (Liu et al., 2024) | Swin-T | OG, Cap4M | 172M | 27.4 | 18.1 | 23.3 | 32.7 | - | - | - | - | **49.7** | 22.3 |
| OVLW-DETR-L (Wang et al., 2024) | LW-DETR-L | OG | 47M | 33.5 | 26.5 | 33.9 | 34.4 | - | - | - | - | - | - |
| OmDet-Turbo-B (Zhao et al., 2024) | ConvNeXt-B | OG | 175M | 34.7 | - | - | - | - | - | - | - | - | - |
| YOLO-Worldv2-L (Cheng et al., 2024) | YOLOv8-L | OG | 48M | 35.4 | 27.6 | 34.1 | 38.0 | 26.8 | 19.8 | 23.6 | 33.4 | 38.4 | 17.1 |
| GDINO 1.5 Edge (Ren et al., 2024) | EfficientViT-L1 | Grounding-20M | - | 33.5 | 28.0 | 34.3 | 33.9 | 27.3 | 26.3 | 25.7 | 29.6 | - | - |
| YOLOE-8-L (Wang et al., 2025) | YOLOv8-L | OG | 45M | 35.9 | 33.2 | 34.8 | 37.3 | - | - | - | - | - | - |
| WeDetect-T (Fu et al., 2025) | ConvNeXt-T | ~20M | 33M | **37.4** | 33.3 | 36.8 | **38.8** | **31.4** | 24.7 | 29.2 | **36.8** | 46.4 | 21.1 |
| ExpAlign (Ours) | ConvNeXt-T | OG | 60M | 37.2 | 35.8 | **37.2** | 37.6 | 30.3 | **26.5** | 29.8 | 33.7 | 48.0 | **22.6** |
| ExpAlign (Ours) | ConvNeXt-T | OG, RefC | 60M | 37.1 | **36.2** | 37.1 | 37.4 | 29.5 | 24.8 | 28.0 | 33.4 | 47.7 | 22.4 |

throughout training. In the first stage, the model is trained for 30 epochs using only the standard YOLOv8 detection and segmentation losses, with an initial learning rate $lr_0 = 0.002$, final learning rate ratio $lrf = 0.01$, and a warmup of 3 epochs. In the second stage, we enable the multi-positive InfoNCE loss and the Geometry-Aware Consistency Objective (GACO), and continue training for another 20 epochs with a reduced initial learning rate $lr_0 = 0.001$, $lrf = 0.2$, and no warmup. The loss weights are fixed to $\lambda_{sem} = 0.5$ and $\lambda_{geo} = 1.0$ across all experiments. The semantic contrastive loss is applied at the lowest feature resolution for efficiency, while GACO is computed on high-resolution alignment maps to preserve spatial structure.

### 4.2. Zero-shot Detection and Segmentation Performance

ExpAlign exhibits competitive zero-shot open-vocabulary detection performance under fair pre-training and inference conditions. We perform zero-shot evaluation on the val and minival splits of LVIS with fixed AP protocol. LVIS features 1203 classes with a long-tail distribution, while ODinW spans 35 diverse real-world datasets, testing generalization to varied domains and vocabularies.

As shown in Table 1, ExpAlign with OG and RefC achieves 37.1 AP on LVIS minival and leads in rare-category performance with AP$_r$ of 36.2. On full LVIS val, it reaches 29.5 AP and 24.8 AP$_r$, benefiting from RefCOCO referring expression supervision for improved long-tail handling. On ODinW, it attains 47.7 AP on ODinW13 and 22.4 AP on ODinW35, substantially outperforming GLIP-T and closely matching or exceeding Grounding DINO-T, using a lightweight design with only 60M total parameters (26M frozen), which enables superior efficiency under comparable backbone scale. WeDetect-Tiny reports 33.3 AP$_r$ on LVIS minival and 24.7 AP$_r$ on LVIS val, along with 46.4 AP on ODinW13 and 21.1 AP on ODinW35, while ExpAlign

*Table 2.* **Zero-shot instance segmentation performance** on LVIS val set using standard mask AP$^m$. ExpAlign and YOLOE are evaluated purely zero-shot without any LVIS images or annotations during training. In contrast, YOLO-Worldv2-L is fine-tuned on LVIS-Base data for the segmentation head.

| Model | AP$^m$ | AP$^m_r$ | AP$^m_c$ | AP$^m_f$ |
|---|---|---|---|---|
| YOLO-Worldv2-L | 19.8 | 17.2 | 17.5 | 23.6 |
| OpenSeeD (Zhang et al., 2023) | 21.0 | - | - | - |
| YOLOE-v8-L | 23.5 | 21.9 | 21.6 | 26.4 |
| YOLOE-11-L | 22.6 | 19.3 | 20.9 | 26.0 |
| ExpAlign | **29.9** | 29.0 | **30.9** | **29.1** |
| ExpAlign ( + RefC) | 29.8 | **29.7** | 30.8 | 28.9 |

achieves superior rare-category performance with 35.8–36.2 AP$_r$ (vs. 33.3) on LVIS minival and 26.5 AP$_r$ (vs. 24.7) on LVIS val, as well as 48.0 AP (vs. 46.4) on ODinW13 and 22.6 AP (vs. 21.1) on ODinW35, while using significantly less pre-training data.

All models use 640×640 input resolution except those with Swin-Tiny backbone, which use 800×1333. These results demonstrate the effectiveness of referring expression data for enhancing rare-object detection and real-world robustness without relying on massive extra pre-training data.

Furthermore, as shown in Table 2, ExpAlign achieves strong zero-shot instance segmentation on the LVIS val set using the standard AP$^m$ metric. It attains 29.9 AP$^m$ overall and 29.0 AP$^m_r$ on rare categories without any exposure to LVIS images during training. This performance far surpasses YOLO-Worldv2-L fine-tuned on LVIS-Base at 19.8 AP$^m$ and YOLOE variants ranging from 22.6 to 23.5 AP$^m$. The substantial improvement of 6 to 10 AP$^m$ is largely attributed to the GACO regularization term introduced during pre-training, which significantly enhances mask precision

*Table 3.* **Downstream fine-tuning performance on COCO.** ExpAlign is fine-tuned on the COCO train2017 set and evaluated on val2017 using standard bounding-box $AP^b$ and mask $AP^m$ metrics, including AP at IoU thresholds 0.50 and 0.75. We compare two practical strategies: linear probing with the backbone frozen for 10 epochs and full tuning with all parameters trainable for 80 epochs. Training-from-scratch baselines are included for reference.

| Model | Epochs | $AP^b$ | $AP^b_{50}$ | $AP^b_{75}$ | $AP^m$ | $AP^m_{50}$ | $AP^m_{75}$ |
|---|---|---|---|---|---|---|---|
| *Training from scratch* | | | | | | | |
| YOLOv8-L | 300 | 52.4 | 69.3 | 57.2 | 42.3 | 66.0 | 44.9 |
| YOLO11-L | 600 | 53.3 | 70.1 | **58.2** | 42.8 | 66.8 | **45.5** |
| *Linear probing* | | | | | | | |
| YOLOE-v8-L | 10 | 45.4 | 63.3 | 50.0 | 38.3 | 59.6 | 40.8 |
| YOLOE-11-L | 10 | 45.1 | 62.8 | 49.5 | 38.0 | 59.2 | 40.6 |
| ExpAlign (Ours) | 10 | 47.2 | 65.3 | 51.7 | 39.2 | 61.7 | 41.8 |
| *Full tuning* | | | | | | | |
| YOLOE-v8-L | 80 | 53.0 | 69.8 | 57.9 | 42.7 | 66.5 | 45.6 |
| YOLOE-11-L | 80 | 52.6 | 69.7 | 57.5 | 42.4 | 66.2 | 45.2 |
| ExpAlign (Ours) | 80 | **53.5** | **70.7** | 57.8 | **42.9** | **67.0** | **45.5** |

*Table 4.* **Performance on common referring expression comprehension** datasets. The evaluation metric for RefCOCO, RefCOCO+, and RefCOCOg is the Top-1 accuracy. * indicates removed mosaic, flip, and HSV augmentations in phase-2 training.

| Method | Pre-Train Data | RefCOCO | | | RefCOCO+ | | | RefCOCOg | |
|---|---|---|---|---|---|---|---|---|---|
| | | val | testA | testB | val | testA | testB | val | test |
| YOLOE-8-L | OG | 32.8 | 43.7 | 45.7 | 28.5 | 31.1 | 45.8 | 22.6 | 23.1 |
| GDINO-T | OG | 50.4 | 57.2 | 43.2 | 51.4 | 57.6 | 45.8 | 67.5 | 67.1 |
| GDINO-T | OG, RefC | 74.0 | 74.9 | 59.3 | 66.8 | 69.9 | 56.1 | 71.1 | 72.1 |
| ExpAlign | OG | 42.2 | 48.2 | 46.7 | 37.62 | 35.3 | 45.2 | 60.0 | 59.67 |
| ExpAlign* | OG, RefC | 51.6 | 59.3 | 47.7 | 48.9 | 47.5 | 45.5 | 65.6 | 64.0 |

*Table 5.* **Ablation study** on token-level alignment versus pooled token representations on the LVIS dataset.

| Alignment Strategy | AP | $AP_r$ | $AP_c$ | $AP_f$ |
|---|---|---|---|---|
| Mean pooling over tokens | 31.9 | 27.3 | 30.5 | 32.2 |
| Global pooled token (EOT) | 34.4 | 33.2 | 34.5 | 35.4 |
| Token-level alignment (EAH) | **37.1** | **36.2** | **37.1** | **37.4** |

and boundary alignment across long-tail categories in open-vocabulary settings.

## 4.3. Downstream Transferring

We evaluate ExpAlign's downstream transferability on the COCO dataset through fine-tuning for object detection and instance segmentation, as shown in Table 3. Under linear probing (backbone frozen, 10 epochs), ExpAlign outperforms YOLOE-v8-L and YOLOE-11-L in both bounding-box $AP^b$ and mask $AP^m$. In full tuning (80 epochs, all parameters trainable), ExpAlign further surpasses these baselines across most metrics, including $AP^b_{50}$, $AP^m_{50}$, and $AP^m_{75}$. These consistent gains across both strategies highlight that ExpAlign's pre-training design enables more efficient and effective adaptation to standard supervised tasks compared to recent open-vocabulary baselines.

## 4.4. Referring Expression Comprehension Performance

As shown in Table 4, ExpAlign underperforms significantly on referring expression comprehension tasks compared to Grounding DINO-T, even when pre-trained with the same RefC data. For example, ExpAlign with RefC achieves only 51.6/59.3/47.7 on RefCOCO splits and 65.6/64.0 on RefCOCOg, far below Grounding DINO-T's 74.0/74.9/59.3 and 71.1/72.1. We acknowledge this limitation openly. The primary cause is likely the CLIP text encoder's inherent weakness in understanding positional and relational language (e.g., "left of", "behind", "next to"), which is crucial for many referring expressions, especially on RefCOCO+ and RefCOCOg. In contrast, Grounding DINO benefits from a more specialized text encoder and fusion mechanism

better suited for spatial reasoning. This highlights a key area for future improvement in ExpAlign's design.

## 4.5. Ablation Study

We further provide extensive analyses for the effectiveness of designs in our ExpAlign. Experiments are conducted on fixed AP (Achal Dave et al., 2022) is reported on LVIS minival splits set for zero-shot evaluation, by default.

As shown in Table 5, the token-level alignment strategy (EAH) significantly outperforms simpler representations. Compared to mean pooling and global pooled token (EOT), EAH improves 5.2 and 2.7 absolute AP points, respectively. More specifically, on rare categories, applying EAH reaches 36.2 $AP_r$, achieving the largest gain of 8.9 $AP_r$ points compared to mean pooling. This demonstrates that explicit alignment at the token level captures finer-grained cross-modal correspondence, leading to better generalization on long-tail distributions.

Table 6 ablates the loss weights for semantic contrastive loss ($\lambda_{sem}$) and geometry-aware consistency ($\lambda_{geo}$). Using $\lambda_{sem} = 0.5$ alone reaches 37.0 AP and strong rare performance (35.8 $AP_r$). Adding $\lambda_{geo}$ (especially at 0.5 or 1.0) consistently improves overall AP and frequent/common categories, with the best results at $\lambda_{sem} = 0.5 + \lambda_{geo} = 0.5$ (37.8 AP) or $\lambda_{sem} = 0.0 + \lambda_{geo} = 1.0$ (37.1 AP, 35.6 $AP_r$). Excessive $\lambda_{sem}$ tends to hurt rare-category performance when combined with high $\lambda_{geo}$, indicating a necessary balance. These results confirm that the geometry-aware term complements semantic alignment by enforcing spatial consistency across the long tail.

Table 7 compares different backbones under identical detec-

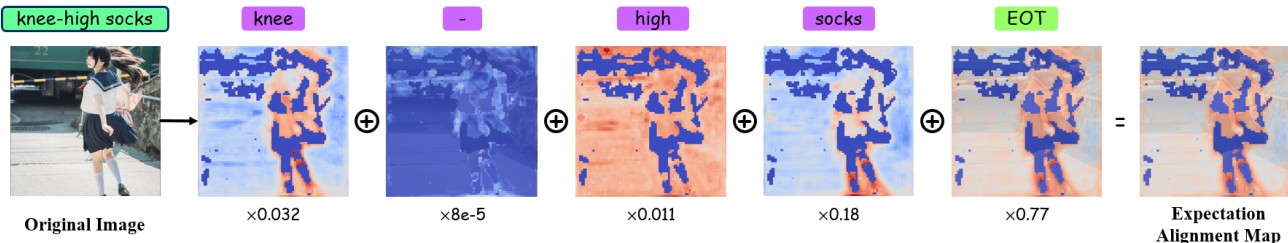

*Figure 2.* Expectation alignment map calculation diagram. Spatial alignment maps are first computed for individual text tokens. All maps are then aggregated with their importance weight (displayed below each map) to form a prompt-conditioned expectation alignment map.

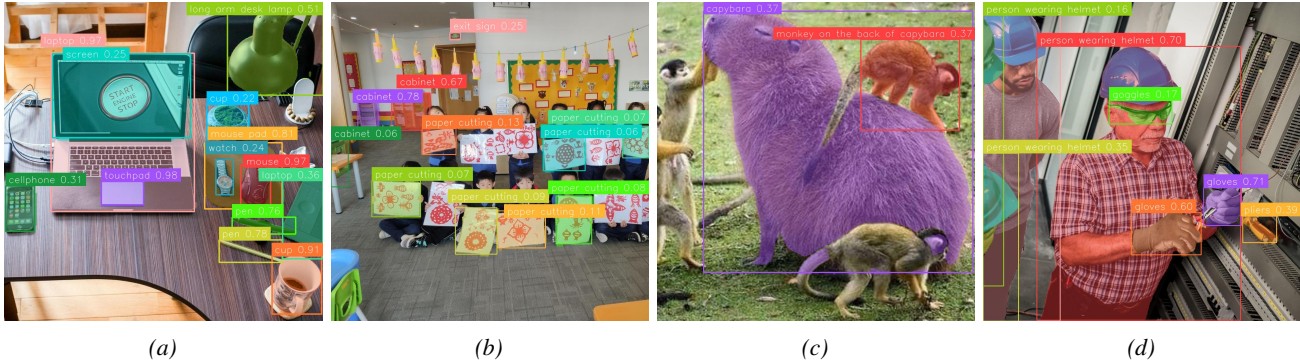

| *(a)* | *(b)* | *(c)* | *(d)* |

*Figure 3.* Qualitative examples of detection and segmentation results. (a) prompts: laptop, cellphone, watch, cup, mouse, long arm desk lamp, pen, mouse pad, touchpad, screen, keyboard. (b) prompts: paper cutting, cabinet, exit sign. (c) prompts: capybara, monkey on the back of capybara. (d) prompts: person wearing helmet, pliers, gloves, goggles. Zoom in for better visual effect.

*Table 6.* **Ablation study on the loss weights** $\lambda_{\text{sem}}$ (semantic contrastive loss) and $\lambda_{\text{geo}}$ (geometry-aware consistency objective) on the LVIS dataset.

| $\lambda_{\text{sem}}$ | $\lambda_{\text{geo}}$ | AP | $\text{AP}_r$ | $\text{AP}_c$ | $\text{AP}_f$ |
|---|---|---|---|---|---|
| 0.0 | 0.0 | 35.0 | 32.0 | 35.8 | 35.1 |
| 0.5 | 0.0 | 37.0 | 35.8 | 35.9 | 38.1 |
| 1.0 | 0.0 | 36.1 | 30.2 | 33.1 | 36.7 |
| 0.0 | 0.5 | 37.6 | 31.7 | **37.7** | **38.6** |
| 0.5 | 0.5 | **37.8** | 32.1 | 37.6 | 38.3 |
| 1.0 | 0.5 | 36.2 | 29.2 | 26.1 | 37.6 |
| 0.0 | 1.0 | 37.1 | 35.6 | 37.6 | 37.2 |
| 0.5 | 1.0 | 37.2 | **35.9** | 37.2 | 37.6 |
| 1.0 | 1.0 | 36.9 | 32.3 | 36.3 | 37.7 |
| 0.0 | 1.5 | 36.6 | 34.5 | 36.2 | 37.4 |
| 0.5 | 1.5 | 36.7 | 35.0 | 36.6 | 37.2 |
| 1.0 | 1.5 | 35.1 | 31.9 | 35.4 | 36.0 |

*Table 7.* **Backbone comparison** on the LVIS dataset. All settings use the same detection and segmentation heads.

| Backbone | AP | $\text{AP}_r$ | $\text{AP}_c$ | $\text{AP}_f$ |
|---|---|---|---|---|
| YOLOv8 | 35.6 | 33.9 | 35.8 | 37.3 |
| DINOv3 | N/A | N/A | N/A | N/A |
| DINOv3 (frozen) | 37.2 | 35.9 | 37.2 | 37.6 |

tion and segmentation heads on LVIS$^{\text{minival}}$. The YOLOv8 backbone achieves 35.6 AP overall, with 33.9 $\text{AP}_r$ on rare categories. In contrast, using the DINOv3 backbone without freezing leads to training collapse (indicated by N/A), resulting in no meaningful convergence. However, when the DINOv3 backbone is frozen during pre-training, ExpAlign reaches 37.2 AP overall and 35.9 $\text{AP}_r$, outperforming YOLOv8 by 1.6 AP and showing particular gains on rare categories. This suggests that preserving the rich, high-quality pre-trained features from a strong frozen vision foundation model is crucial for ExpAlign's cross-modal alignment objectives, whereas a detection-oriented backbone like YOLOv8 or unfrozen DINOv3 hinders effective learning of the alignment signals.

## 5. Visualization

Figure 2 illustrates the intuition behind the proposed EAH. Instead of collapsing a text prompt into a single global embedding, EAH preserves all token-level representations before the EOT token and computes a spatial alignment map for each token. These token-wise maps are then combined through a soft expectation mechanism, where each token contributes with a learned importance weight. As a result, the EOT token remains the dominant alignment signal inherited from CLIP pre-training, while informative non-EOT tokens (e.g., *knee*, *high*, *socks*) provide complementary fine-grained cues that refine the spatial structure of the alignment

map. Rather than suppressing the EOT token, Expectation Alignment enhances it with token-level semantic details, enabling fine-grained vision-language alignment without introducing hard token selection or additional supervision.

We present qualitative results of ExpAlign in Figure 3. Subfigures 3a and 3b present correctly detected objects, where all corresponding prompt phrases are completely absent from the training data. This clearly demonstrates the robust zero-shot generalization capability of ExpAlign. Furthermore, the results in subfigures 3c and 3d reveal that the model exhibits a non-trivial level of referring expression comprehension (REC) ability, successfully grounding complex and novel expressions even in unseen scenarios. Notably, ExpAlign delivers exceptionally high-quality segmentation masks, with particularly impressive performance at object boundaries and under partial occlusion, highlighting its strong capability in precise instance delineation. See more examples in Appendix H.

## 6. Conclusion

In this paper, we presented **ExpAlign**, an expectation-guided vision-language alignment framework for open-vocabulary grounding under weak and ambiguous supervision. By introducing the Expectation Alignment Head (EAH), our method aggregates token-level vision-language similarities through a principled expectation mechanism, enabling implicit token selection and soft region alignment without relying on explicit instance-level annotations. Furthermore, we proposed a multi-scale consistency regularization strategy, including a Top-K multi-positive contrastive objective and a Geometry-Aware Consistency Objective, to jointly enhance semantic discriminability and spatial coherence of alignment maps during training. Extensive experiments on open-vocabulary detection and instance segmentation benchmarks demonstrate that ExpAlign consistently improves performance, particularly on long-tail categories and zero-shot segmentation quality, while remaining lightweight and fully compatible with standard detection and segmentation pipelines. We believe this work offers a practical and theoretically grounded step toward more expressive and robust vision-language alignment for open-world visual understanding.

## Acknowledgement

This work was supported by Natural Science Foundation of Sichuan Province of China (Grant No. 2025ZNSFSC0522) and partially supported by National Natural Science Foundation of China (Grant No.61571096).

## Impact Statement

This paper presents work whose goal is to advance the field of machine learning. There are many potential societal consequences of our work, none of which we feel must be specifically highlighted here.

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

# A. Connection to Multiple Instance Learning

Although ExpAlign is presented as a vision-language alignment module rather than a canonical MIL algorithm, it admits an exact interpretation and equivalence to attention-based soft pooling in the MIL framework. Below we give a concise mapping and a proof sketch that justifies the claim in Section 3.5.

**Notation.** Fix a textual prompt $p$ and a feature scale $s$. Let $\Omega = \{1, \ldots, N\}$ index spatial locations in the feature map ($N = H_s W_s$), and let $L$ denote the number of valid text tokens. For each spatial location $i \in \Omega$ and token $l \in \{1, \ldots, L\}$, define the token–patch affinity

$$S(i, l) = \langle F_s(x, y), T_{b,p}(l) \rangle,$$

where $i$ is the flattened index of $(x, y)$. The spatially averaged response of token $l$ is

$$\bar{S}(l) = \frac{1}{N} \sum_{i \in \Omega} S(i, l),$$

and the token posterior is given by

$$\pi(l) = \frac{\exp(\bar{S}(l)/\tau_t)}{\sum_{l'=1}^{L} \exp(\bar{S}(l')/\tau_t)}.$$

The EAM assigns to each spatial location the score

$$\widetilde{S}(i) = \sum_{l=1}^{L} \pi(l)\, S(i, l).$$

**Reformulation as instance-wise linear pooling.** For each instance $i$, define the token-affinity vector

$$v_i = (S(i, 1), \ldots, S(i, L))^\top \in \mathbb{R}^L,$$

and collect the token posteriors into $\pi \in \mathbb{R}^L$. With this notation, the EAM score can be written compactly as

$$\widetilde{S}(i) = \pi^\top v_i,$$

which shows that each instance score is obtained by applying the same linear functional to its token-affinity vector.

**MIL interpretation and bag-level aggregation.** From the MIL perspective, the prompt $p$ defines a bag whose instances are the unordered set $\{v_i\}_{i \in \Omega}$. The mapping $v_i \mapsto \widetilde{S}(i)$ is permutation equivariant, and the subsequent aggregation used by ExpAlign,

$$\ell = \frac{1}{|\mathrm{TopK}(\widetilde{S})|} \sum_{i \in \mathrm{TopK}(\widetilde{S})} \widetilde{S}(i),$$

is permutation invariant. Such a construction satisfies the defining requirement of MIL pooling operators and corresponds to a Top-$K$ variant of attention-based soft pooling, where discriminative instances dominate the bag-level response.

**Equivalence to attention-based MIL pooling.** Attention-based MIL methods (Ilse et al., 2018) compute a scalar score for each instance via an attention mechanism and aggregate these scores using a permutation-invariant operator. In ExpAlign, attention is factorized into a token-level posterior $\pi$, shared across instances, followed by instance-level pooling over $\widetilde{S}(i)$. Algebraically, both formulations reduce to computing instance scores $g(v_i)$ and applying a soft or Top-$K$ aggregation over instances. The difference lies only in how attention weights are parameterized, not in the form of the pooling operator.

**Permutation invariance and expressiveness.** Because $\pi$ depends only on the set $\{v_i\}$ through the averaged statistics $\{\bar{S}(l)\}$, the overall operator from $\{v_i\}$ to $\ell$ is permutation invariant. Moreover, by adjusting the temperature $\tau_t$ and the Top-$K$ ratio, the pooling behavior interpolates between mean, max, and soft-attention pooling, matching the expressive family of attention-based MIL operators.

**Discussion.** This equivalence clarifies that ExpAlign performs a principled MIL-style soft selection over instances while allowing uncertainty at both the token and spatial levels. The auxiliary multi-positive InfoNCE loss and the Geometry-Aware Consistency Objective can thus be viewed as bag-level discriminative losses and intra-bag energy shaping, respectively, consistent with standard MIL training principles.

**Remarks.** For completeness, the variational derivation in Appendix B further shows that the geometry-aware consistency term yields a Gibbs reweighting of prompt–patch probabilities under a Lagrangian-constrained free-energy, which reshapes intra-instance mass without requiring explicit instance labels.

## B. Variational Derivation of Gibbs Reweighting in Energy-Based Consistency Regularization

We consider a finite collection of prompt–patch pairs indexed by $(p, i)$, with $p \in \{1, \dots, P\}$ and $i \in \Omega$ ($|\Omega|$ finite). For compactness we sometimes write a generic index $\alpha$ to denote a pair $(p, i)$.

**Assumption B.1** (Energy field). There is a real-valued alignment score field $\tilde{S}_p(i) \in \mathbb{R}$. We define the associated *energy* by

$$E(p, i) = -\tilde{S}_p(i). \tag{16}$$

We assume $E(p, i)$ is uniformly bounded on the finite domain.

**Assumption B.2** (Instance-wise geometry score). For each image $b$ and prompt $p$ we are given a bounded geometry score $A_{b,p}(i) \in \mathbb{R}$ defined for $i \in \Omega$ such that:

1. $A_{b,p}$ depends only on intra-instance relative statistics (e.g. mean and standard deviation computed over the ground-truth mask $\mathcal{M}_{b,p}$), and is therefore invariant to adding a constant to $\tilde{S}$ (affine invariance in the additive sense) and to monotone affine rescaling when appropriately adjusting normalization;

2. $A_{b,p}$ is bounded and (locally) Lipschitz in $\tilde{S}$ (so gradient bounds exist and empirical gradients are well-defined).

**Assumption B.3** (Regularization parameters). Let $\tau > 0$ be the temperature (entropy weight) and $\lambda \in \mathbb{R}$ be the geometry weight (we will take $\lambda \geq 0$ in most discussion).

We denote by $\mathcal{P}$ the probability simplex over all prompt–patch indices:

$$\mathcal{P} = \left\{ \mathbb{Q} : \mathbb{Q}(p, i) \geq 0, \ \sum_{p,i} \mathbb{Q}(p, i) = 1 \right\}.$$

Let $\mathbb{U}$ denote the uniform distribution on the finite set of pairs $(p, i)$, i.e. $\mathbb{U}(p, i) = 1/(P|\Omega|)$.

**Theorem B.4** (Variational optimality and induced Gibbs form). *Under Assumptions B1–B3, consider the variational free-energy functional*

$$\mathcal{F}[\mathbb{Q}] = \mathbb{E}_{\mathbb{Q}}\big[E(p, i)\big] - \lambda \mathbb{E}_{\mathbb{Q}}\big[A_{b,p}(i)\big] + \tau \, \mathrm{KL}\big(\mathbb{Q} \, \| \, \mathbb{U}\big), \qquad \mathbb{Q} \in \mathcal{P}. \tag{17}$$

*Then:*

1. *The functional $\mathcal{F}$ is strictly convex on $\mathcal{P}$ and admits a unique minimizer $\mathbb{Q}^\star \in \mathcal{P}$.*

2. *The minimizer has the explicit Gibbs (exponential-family) form*

$$\mathbb{Q}^\star(p, i) = \frac{\exp\big(-\frac{1}{\tau}\big(E(p, i) - \lambda A_{b,p}(i)\big)\big)}{\sum_{p',i'} \exp\big(-\frac{1}{\tau}\big(E(p', i') - \lambda A_{b',p'}(i')\big)\big)}. \tag{18}$$

3. *Equivalently, substituting $E(p, i) = -\tilde{S}_p(i)$, the optimal distribution can be written*

$$\mathbb{Q}^\star(p, i) = \frac{\exp\big(\frac{1}{\tau}\big(\tilde{S}_p(i) + \lambda A_{b,p}(i)\big)\big)}{\sum_{p',i'} \exp\big(\frac{1}{\tau}\big(\tilde{S}_{p'}(i') + \lambda A_{b',p'}(i')\big)\big)}.$$

4. *Moreover, minimizing the cross-entropy (or KL divergence) from an empirical target distribution $Q_{\text{target}}(p, i) \propto \mathbb{K}_{i \in \mathcal{M}_{b,p}} w_{b,p}(i)$ to $\mathbb{Q}^\star$ yields the geometry-aware loss of the form*

$$\mathcal{L}_{\text{geo}} = -\sum_{b,p} \sum_{i \in \mathcal{M}_{b,p}} \tilde{c}_{b,p,i} \log \mathbb{Q}^\star(p, i),$$

*which is equivalent up to normalization constants to the loss reported in the main text.*

**Remarks.**

- The KL term provides strict convexity and enforces positive entropy, preventing collapse to a point mass; the linear terms (expectation of $E$ and $A$) are affine in $\mathbb{Q}$ and therefore preserve convexity.

- The parameter $\tau$ controls the trade-off between fidelity to energy $E$ and entropy (stability vs. selectivity); $\lambda$ controls the strength of instance-local geometric shaping.

- Additive shifts of $E$ (i.e. $E \mapsto E + c$) do not change $\mathbb{Q}^\star$; multiplicative rescaling of $E$ can be absorbed into $\tau$ (i.e. $aE/\tau = (E)/(\tau/a)$).

*Proof.* We supply a complete and explicit derivation in several carefully enumerated steps.

Work on the finite index set $\mathcal{I} = \{(p, i)\}$. Any $\mathbb{Q} \in \mathcal{P}$ can be represented as a vector $\mathbb{Q} \in \mathbb{R}^{|\mathcal{I}|}$ with nonnegative entries summing to one. On this finite dimensional simplex all functions below are well-defined and differentiable on the interior.

**Convexity and existence/uniqueness.**

Observe that $\mathcal{F}[\mathbb{Q}]$ in (17) can be written as

$$\mathcal{F}[\mathbb{Q}] = \sum_{(p,i)\in\mathcal{I}} \mathbb{Q}(p,i)\big(E(p,i) - \lambda A_{b,p}(i)\big) + \tau \sum_{(p,i)} \mathbb{Q}(p,i) \log \frac{\mathbb{Q}(p,i)}{\mathbb{U}(p,i)}.$$

The first term is linear in $\mathbb{Q}$; the second term is $\tau$ times the relative entropy (KL), which is strictly convex in $\mathbb{Q}$ on the interior of the simplex. Hence $\mathcal{F}$ is strictly convex. Because $\mathcal{P}$ is compact and $\mathcal{F}$ is continuous, a unique minimizer exists.

**First-order optimality (variational derivative).**

To find the minimizer, form the Lagrangian for the constrained minimization (constraint: $\sum_{(p,i)} \mathbb{Q}(p,i) = 1$):

$$\mathcal{L}(\mathbb{Q}, \eta) = \sum_{(p,i)} \mathbb{Q}(p,i)\big(E(p,i) - \lambda A_{b,p}(i)\big) + \tau \sum_{(p,i)} \mathbb{Q}(p,i) \log \frac{\mathbb{Q}(p,i)}{\mathbb{U}(p,i)} + \eta\Big(\sum_{(p,i)} \mathbb{Q}(p,i) - 1\Big),$$

where $\eta \in \mathbb{R}$ is the Lagrange multiplier enforcing normalization.

Take partial derivative with respect to $\mathbb{Q}(\bar{p}, \bar{i})$ (interior point) and set to zero:

$$0 = \frac{\partial \mathcal{L}}{\partial \mathbb{Q}(\bar{p}, \bar{i})} = E(\bar{p}, \bar{i}) - \lambda A_{b,\bar{p}}(\bar{i}) + \tau\Big(\log \frac{\mathbb{Q}(\bar{p}, \bar{i})}{\mathbb{U}(\bar{p}, \bar{i})} + 1\Big) + \eta.$$

Rearrange to isolate the log term:

$$\log \frac{\mathbb{Q}(\bar{p}, \bar{i})}{\mathbb{U}(\bar{p}, \bar{i})} = -\frac{1}{\tau}\big(E(\bar{p}, \bar{i}) - \lambda A_{b,\bar{p}}(\bar{i})\big) - 1 - \frac{\eta}{\tau}.$$

Exponentiating both sides yields

$$\mathbb{Q}(\bar{p}, \bar{i}) = \mathbb{U}(\bar{p}, \bar{i}) \exp\Big(-\frac{1}{\tau}\big(E(\bar{p}, \bar{i}) - \lambda A_{b,\bar{p}}(\bar{i})\big)\Big) \cdot \exp\Big(-1 - \frac{\eta}{\tau}\Big).$$

Since $\exp(-1 - \eta/\tau)$ is a global scalar independent of $(\bar{p}, \bar{i})$, normalization enforces that this scalar equals the reciprocal of the partition sum. Using the explicit form of $\mathbb{U}(p, i)$ (uniform), we obtain the normalized Gibbs form

$$\mathbb{Q}^\star(p, i) = \frac{\exp\big(-\frac{1}{\tau}\big(E(p,i) - \lambda A_{b,p}(i)\big)\big)}{\sum_{p',i'} \exp\big(-\frac{1}{\tau}\big(E(p',i') - \lambda A_{b',p'}(i')\big)\big)}.$$

This completes the derivation of (18) and establishes both necessity and sufficiency of this form for optimality (sufficiency follows from strict convexity).

**Substitution $E = -\tilde{S}$ and alternative form.** Using $E(p,i) = -\tilde{S}_p(i)$, rewrite (18) as

$$\mathbb{Q}^{\star}(p,i) = \frac{\exp\left(\frac{1}{\tau}\left(\tilde{S}_p(i) + \lambda A_{b,p}(i)\right)\right)}{\sum_{p',i'} \exp\left(\frac{1}{\tau}\left(\tilde{S}_{p'}(i') + \lambda A_{b',p'}(i')\right)\right)}.$$

This shows that the geometry term $A_{b,p}(i)$ directly enters the logits of the Gibbs distribution and hence modifies the model posterior in a multiplicative exponential manner.

**Equivalence to cross-entropy style training loss.**

Suppose we define a target empirical distribution on $(p,i)$ for training,

$$Q_{\text{target}}(p,i) = \frac{\mathbb{K}\{i \in \mathcal{M}_{b,p}\}\, w_{b,p}(i)}{\sum_{p',i'} \mathbb{K}\{i' \in \mathcal{M}_{b',p'}\}\, w_{b',p'}(i')},$$

where $w_{b,p}(i)$ is a nonnegative weight (e.g. $w_{b,p}(i) = A_{b,p}(i)$ or another monotone transform). The standard cross-entropy (expected negative log-likelihood) of this target under model $\mathbb{Q}$ is

$$\mathrm{CE}(Q_{\text{target}}\|\mathbb{Q}) = -\sum_{p,i} Q_{\text{target}}(p,i)\, \log \mathbb{Q}(p,i).$$

Minimizing this CE over model parameters (i.e. making $\mathbb{Q}$ approximate $Q_{\text{target}}$) is equivalent to minimizing $\mathrm{KL}(Q_{\text{target}}\|\mathbb{Q})$ up to an additive entropy constant $H(Q_{\text{target}})$ independent of model. When the model is constrained to the Gibbs family as in (18), minimizing CE corresponds to adjusting free-energy parameters (and indirectly logits $\tilde{S}$ and geometry weight $\lambda$) so that $\mathbb{Q}^{\star}$ matches $Q_{\text{target}}$. Thus the training objective

$$\mathcal{L}_{\text{geo}} = -\sum_{b,p} \sum_{i \in \mathcal{M}_{b,p}} \tilde{c}_{b,p,i}\, \log \mathbb{Q}^{\star}(p,i)$$

is precisely the empirical counterpart of the variational optimization (17) when choosing $Q_{\text{target}}$ proportional to instance-local geometry weights. $\qquad\square$

**Additional properties (invariance and non-collapse).**

- *Additive invariance.* If $E \mapsto E + c$ (for constant $c$), then the numerator of (18) acquires factor $\exp(-c/\tau)$ independent of $(p,i)$ and cancels with the denominator; hence $\mathbb{Q}^{\star}$ is invariant to additive shifts of energy, equivalently to adding constants to $\tilde{S}$.

- *Scaling and temperature.* If $E$ is multiplied by positive scalar $a > 0$, then

$$\exp\left(-\tfrac{1}{\tau}aE\right) = \exp\left(-\tfrac{1}{\tau/a}E\right),$$

so multiplicative rescaling of $E$ can be absorbed into a reparametrization of $\tau$ (temperature).

- *Non-collapse (positive entropy).* Because $\tau > 0$ and the KL term penalizes zero entropy, the minimizer $\mathbb{Q}^{\star}$ has strictly positive entropy (unless the energy differences are arbitrarily large compared to $\tau$). In particular $\mathbb{Q}^{\star}$ is not a point mass unless the limit $\tau \downarrow 0$ is taken.

- *Instance-local perturbation.* If $A_{b,p}(i)$ is supported only on indices $i$ belonging to ground-truth instance $\mathcal{M}_{b,p}$, then the additive perturbation $\lambda A_{b,p}(i)$ only affects relative probabilities within that instance; it does not change ordering of energies between different instances except insofar as their partition sums change, and thus constitutes a *conditional* (instance-wise) energy shaping.

**Limit cases and interpretation.**

- As $\tau \to \infty$, the KL penalty dominates and $\mathbb{Q}^{\star}$ tends to the uniform distribution $\mathbb{U}$ (max-entropy limit).

- As $\tau \to 0^+$, $\mathbb{Q}^\star$ concentrates on the minimizers of $E(p, i) - \lambda A_{b,p}(i)$ (hard selection / argmax).

- As $\lambda \to 0$, one recovers the standard Gibbs posterior based solely on $E$ (i.e. the semantic-only reweighting).

- Intermediate $(\tau, \lambda)$ trade off stability (entropy), semantic fidelity (alignment to $\tilde{S}$), and geometric consistency.

This completes the derivation and justification of the Gibbs reweighting and conditional energy shaping regularizers used in the main text.

## C. Comparative Evaluation of ExpAlign, Grounding DINO, and GLIP on Diverse Real-World Datasets in ODinW

In our comparison of ExpAlign, Grounding DINO, and GLIP across the diverse real-world datasets in the ODinW benchmark, as presented in Table 8, ExpAlign demonstrates competitive overall performance with a slightly higher average score (22.4) than Grounding DINO (22.3) and a clear advantage over GLIP (19.6), while also showing strong gains on several challenging and domain-specific subsets.

For instance, ExpAlign substantially outperforms both baselines on datasets involving uncommon or underrepresented scenarios, such as MountainDewCommercial (45.46 vs. 25.46 for Grounding DINO and 21.60 for GLIP), ShellfishOpenImages* (42.63 vs. 29.56 and 25.90), MaskWearing (7.83 vs. 0.25 and 1.10), and PKLot_640 (5.23 vs. 0.06 and 0.00). These improvements are likely attributable to ExpAlign's training on RefCOCO, which emphasizes referring expression comprehension and finer-grained grounding of objects in complex or natural-language contexts, helping the model better handle rare categories, occluded objects, or domain shifts not well covered in the O365 + GoldG + Cap4M pre-training corpus shared by Grounding DINO and GLIP.

Notably, on the ODinW-13 benchmark subset (marked with *), ExpAlign also achieves leading results in several cases (e.g., CottontailRabbits, EgoHands_generic, pistols, VehiclesOpenImages), underscoring its enhanced generalization in high-quality, diverse open-world evaluation settings. These observations highlight the value of incorporating referring expression data during pre-training to boost robustness on out-of-distribution and long-tail categories in real-world object detection.

## D. GACO Pseudocode

This section provides the pseudocode of the Geometry-Aware Consistency Objective (GACO) used in all experiments. The objective operates on prompt-conditioned patch-level similarity maps and enforces consistency by reshaping the distribution of alignment scores within positive regions. Specifically, GACO treats the normalized similarity scores as an energy field over spatial locations and derives a Gibbs-style reweighting through a log-softmax normalization. Within each positive region, instance-level responses are standardized using region statistics, yielding an advantage signal that emphasizes relatively confident locations while preserving uncertainty. The final loss is computed as an advantage-weighted log-likelihood over masked locations, which corresponds to a constrained reweighting of spatial energies rather than explicit instance-level supervision. Algorithm 1 summarizes the exact implementation used in our method.

## E. Hyper Paramerter Settings

ExpAlign is trained using a two-stage protocol with frozen image and text encoders in both stages. Stage 1 focuses on semantic alignment with a moderate learning rate schedule and standard augmentations, while Stage 2 introduces geometry-aware consistency (GACO) and multi-positive contrastive weighting with reduced augmentation strength. Detailed hyperparameters are provided in Table 9.

## F. EAM Heatmap Visualizations for Negative Prompts

Figure 4 presents the visualization of Explainable Attention Map (EAM) for negative sample prompt words on an example image of a girl in a sailor uniform. The left subfigure shows the original image, while the middle and right subfigures display EAM heatmaps for the positive prompt *sailor uniform* and negative prompt *black sailor uniform*, respectively.

As observed, for negative prompt, the EAM activations are more uniformly distributed across the background rather than

| Metric | GLIP-T | Grounding DINO T | ExpAlign |
|---|---|---|---|
| Average Score | 19.6 | 22.3 | **22.4** |
| Median Score | 5.1 | **11.9** | 10.4 |
| AerialMaritimeDrone_large* | **13.70** | 10.30 | 11.15 |
| AerialMaritimeDrone_tiled | 12.60 | 17.50 | **23.66** |
| AmericanSignLanguageLetters | **2.50** | 0.78 | 2.34 |
| Aquarium* | 18.30 | 18.64 | **19.88** |
| BCCD | 1.00 | 11.96 | **13.98** |
| ChessPieces | 10.00 | **15.62** | 7.87 |
| CottontailRabbits* | 69.70 | 67.61 | **81.34** |
| DroneControl | **5.10** | 4.99 | 0.37 |
| EgoHands_generic* | 50.00 | 57.64 | **61.50** |
| EgoHands_specific | 0.80 | 0.69 | **0.91** |
| HardHatWorkers | 3.00 | **4.05** | 3.30 |
| MaskWearing | 1.10 | 0.25 | **7.83** |
| MountainDewCommercial | 21.60 | 25.46 | **45.46** |
| NorthAmericaMushrooms* | **75.10** | 68.18 | 25.38 |
| OxfordPets_by-breed | 0.40 | 0.21 | **2.01** |
| OxfordPets_by-species | 1.10 | 1.30 | **7.20** |
| PKLot_640 | 0.00 | 0.06 | **5.23** |
| Packages* | **72.30** | 60.53 | 71.65 |
| PascalVOC* | 56.10 | 55.65 | **59.00** |
| Raccoon* | 57.80 | **60.07** | 46.29 |
| ShellfishOpenImages* | 25.90 | 29.56 | **42.63** |
| ThermalCheetah | 2.70 | **17.72** | 12.65 |
| UnoCards | 0.20 | 0.81 | **0.99** |
| VehiclesOpenImages* | 56.00 | 58.49 | **61.40** |
| WildfireSmoke | 2.30 | **20.04** | 10.38 |
| boggleBoards | 0.00 | 0.29 | **1.33** |
| brackishUnderwater | 3.70 | 1.47 | **3.70** |
| dice | **1.10** | 0.33 | 0.36 |
| openPoetryVision | 0.00 | 0.05 | **0.16** |
| pistols* | 49.80 | 66.99 | **75.69** |
| plantdoc | 1.10 | 0.36 | **2.37** |
| pothole* | 17.20 | **25.21** | 7.30 |
| selfdrivingCar | 8.00 | **9.95** | 8.21 |
| thermalDogsAndPeople* | 43.70 | **67.89** | 57.04 |
| websiteScreenshots | 0.50 | 1.30 | **2.64** |

*Table 8.* Comparison of ExpAlign, Grounding DINO, and GLIP on the ODinW benchmark. Grounding DINO and GLIP are trained on Objects365, GoldG, and Cap4M using Swin-Tiny backbones.ExpAlign is trained on Objects365, GoldG, and RefCOCO using ConvNeXt-Tiny backbones. *denotes results belonging to the ODinW-13 benchmark.

---

**Algorithm 1** Geometry-Aware Consistency Objective (GACO)

---

**Input:** similarity map $sim \in [-1, 1]$ of shape $[B, K, H, W]$, binary mask $M \in \{0, 1\}$ of shape $[B, K, H, W]$, hyperparameters $\beta$, $adv\_clip$, $\epsilon$

**Output:** geometry consistency loss $\mathcal{L}_{\text{geo}}$

Normalize $sim \leftarrow sim/(|sim|_{\max} + \epsilon)$

Compute logits $\leftarrow sim.view(B, K \cdot H \cdot W)$

Compute $\log p \leftarrow \log \text{softmax}(\text{logits}, \dim = 1)$

Flatten $M$ to $M_{\text{flat}} \in [B, K \cdot H \cdot W]$

Compute probability $prob_{\text{flat}} \leftarrow \sigma(sim).view(B, K \cdot H \cdot W)$

Initialize advantage-weighted loss $L_{\text{adv}} \leftarrow 0$, denominator $denom \leftarrow 0$

**for** each batch $b = 0$ **to** $B - 1$ **do**

    Find positive pixel indices $pos\_idx \leftarrow$ where $M_{\text{flat}}[b] > 0.5$

    **if** $pos\_idx$ is not empty **then**

        $R_{\text{pos}} \leftarrow prob_{\text{flat}}[b, pos\_idx]$

        $\mu \leftarrow \text{mean}(R_{\text{pos}}), \sigma \leftarrow \text{std}(R_{\text{pos}}) + \epsilon$

        Advantage $A \leftarrow (R_{\text{pos}} - \mu)/\sigma$

        Clamp $A \leftarrow \text{clamp}(A, -adv\_clip, adv\_clip)$

        Accumulate $L_{\text{adv}} \leftarrow L_{\text{adv}} - (A \cdot \log p[b, pos\_idx]).sum()$

        $denom \leftarrow denom + |pos\_idx|$

    **end if**

**end for**

$L_{\text{adv}} \leftarrow L_{\text{adv}}/denom$ if $denom > 0$ else $0$

$\mathcal{L}_{\text{geo}} \leftarrow \beta \cdot L_{\text{adv}}$

---

| Hyper Paramerter | Stage 1 Training | Stage 2 Training |
|---|---|---|
| image size | 640x640 | 640x640 |
| batch size | 512 | 512 |
| epochs | 30 | 20 |
| warmup epochs | 3 | 0 |
| weight decay | 0.025 | 0.025 |
| initial learning rate | 0.002 | 0.01 |
| final learning rate fraction | 0.001 | 0.1 |
| bias learning rate warmup | 0.0 | 0.0 |
| momentum | 0.9 | 0.9 |
| AMP training | True | True |
| freeze image encoder | True | True |
| freeze text encoder | True | True |
| multi-positive InfoNCE weight | 0.0 | 0.5 |
| GACO weight | 0.0 | 1.0 |
| mosaic | 1.0 | 1.0 |
| close mosaic | 2 | 5 |
| hsv h | 0.015 | 0.005 |
| hsv s | 0.7 | 0.05 |
| hsv v | 0.4 | 0.05 |
| fliplr | 0.5 | 0.0 |

*Table 9.* ExpAlign training hyper-parameters.

concentrating on the foreground object (the uniform). This pattern suggests that the model suppresses the detection of negative prompts by diffusing attention, reducing false positives in irrelevant regions and enhancing overall robustness in prompt-guided tasks.

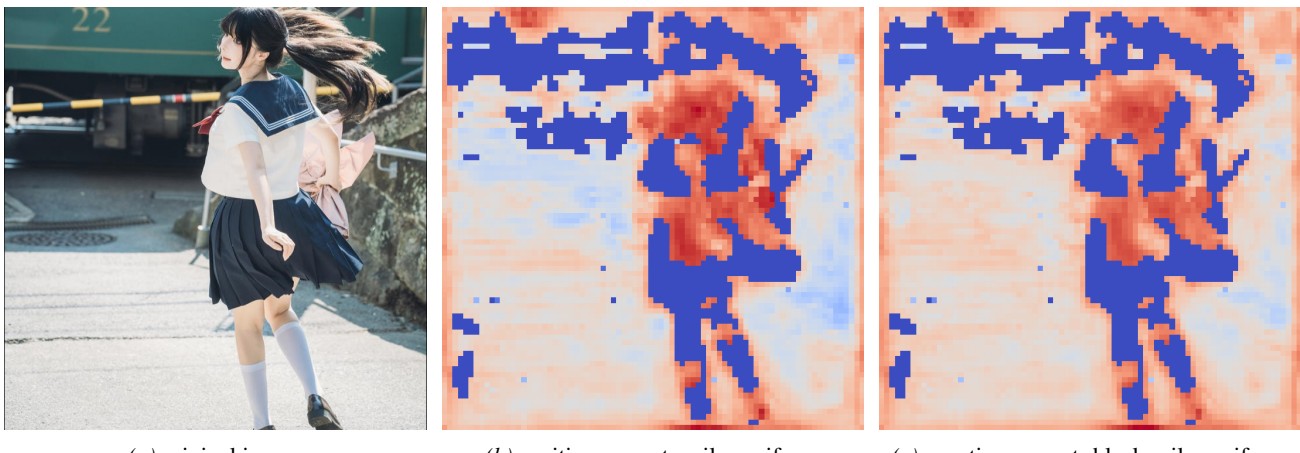

*(a)* original image      *(b)* positive prompt: sailor uniform      *(c)* negative prompt: black sailor uniform

*Figure 4.* EAM heatmaps for positive prompt *sailor uniform* and negative prompt *black sailor uniform*. Background-dominant activations indicate effective suppression of unseen negative prompts.

## G. Impact of Global Negative Vocabulary

During training, we observe that the composition and quality of the global negative vocabulary have a noticeable impact on performance, particularly for rare categories ($AP_r$). Varying the negative prompt set—through different sampling strategies, vocabulary sizes, or semantic distributions—results in fluctuations of approximately $\pm 0.8\%$ in $AP_r$ on the LVIS minival split. In contrast, the effect on overall AP as well as $AP_c$ and $AP_f$ is relatively limited, with variations within $\pm 0.2\%$.

This behavior suggests that rare-category representations in the CLIP embedding space are inherently more fragile and sensitive to interference from negative prompts. When negative samples are semantically close to rare positives or occupy nearby regions in the embedding space, they can induce stronger gradient conflicts during contrastive alignment, disproportionately impairing the model's ability to discriminate long-tail classes. Frequent and common categories, which are more densely covered during vision–language pre-training, exhibit greater robustness to such perturbations.

We hypothesize that an effective negative vocabulary should occupy a "sweet spot" in the CLIP feature space: sufficiently separated from the positive (LVIS) distribution to suppress false activations, yet not so distant that the negatives become uninformative and yield weak or noisy gradients. Negative sets that are overly similar to positives may lead to excessive suppression and hinder rare-category learning, while overly distant negatives may fail to provide meaningful discriminative supervision. Identifying such a balanced negative distribution could further improve performance on LVIS, particularly for long-tail categories.

At present, however, there is no standardized metric or principled methodology to quantify the "quality" or "difficulty" of a global negative vocabulary in open-vocabulary detection. Developing reliable criteria or adaptive strategies for negative vocabulary construction—such as embedding-aware sampling, online hard-negative mining, or dynamic vocabulary curation—remains an open challenge and a promising direction for future work.

## H. More Visualization Examples

Figure 5 and 6 shows additional zero-shot detection and segmentation results of ExpAlign on diverse scenes with multi-object and detailed text prompts. The model demonstrates strong open-vocabulary grounding and precise instance masks across novel categories and complex compositions.

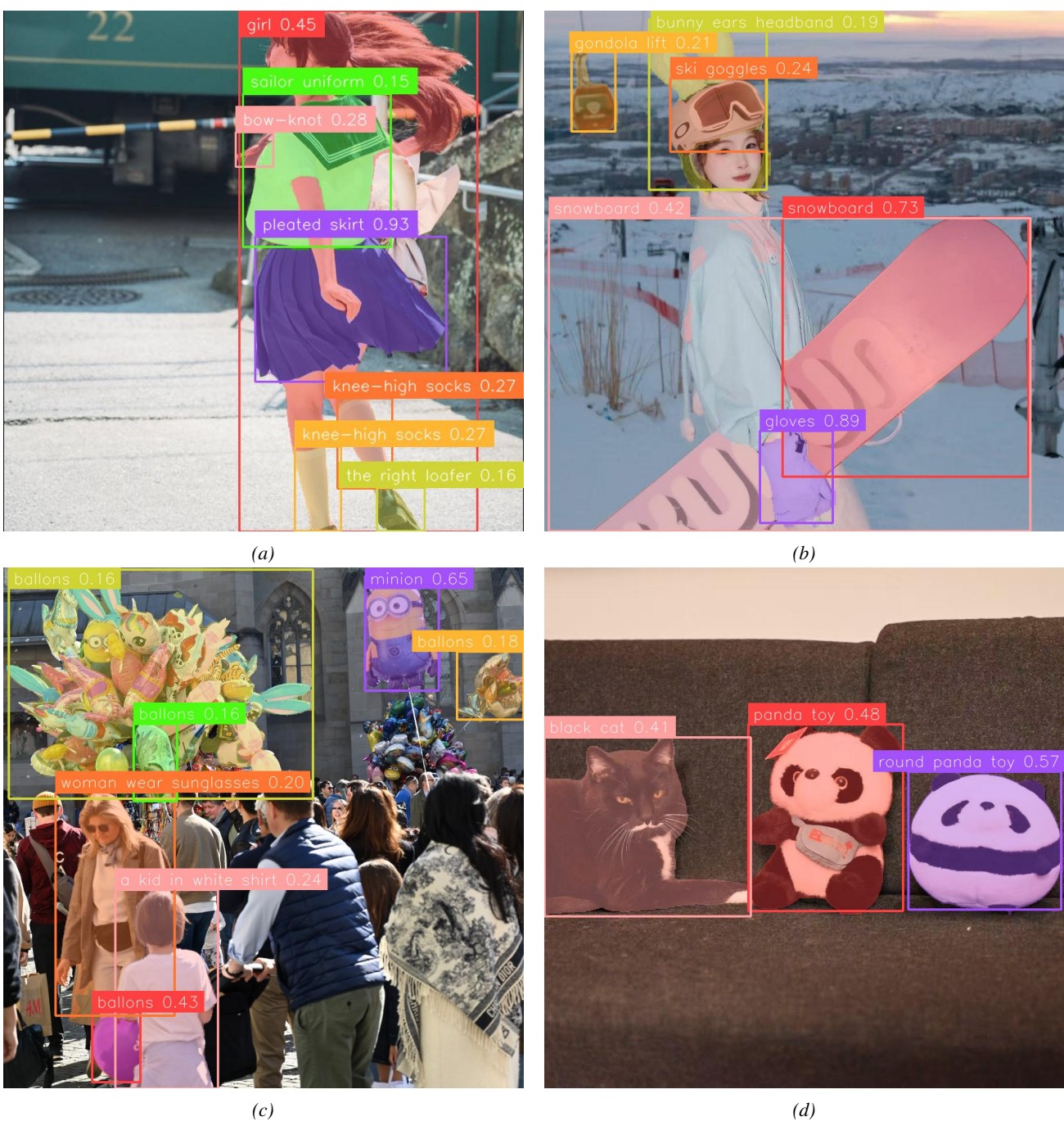

*Figure 5.* (a) prompts: girl, sailor uniform, the right loafer, bow-knot, knee-high socks, pleated skirt. (b) prompts: snowboard, ski goggles, gondola lift, gloves, bunny ears headband. (c) prompts: minion, ballons, a kid in white shirt, woman wear sunglasses. (d) prompts: black cat, panda toy, round panda toy. Zoom in for better visual effect.

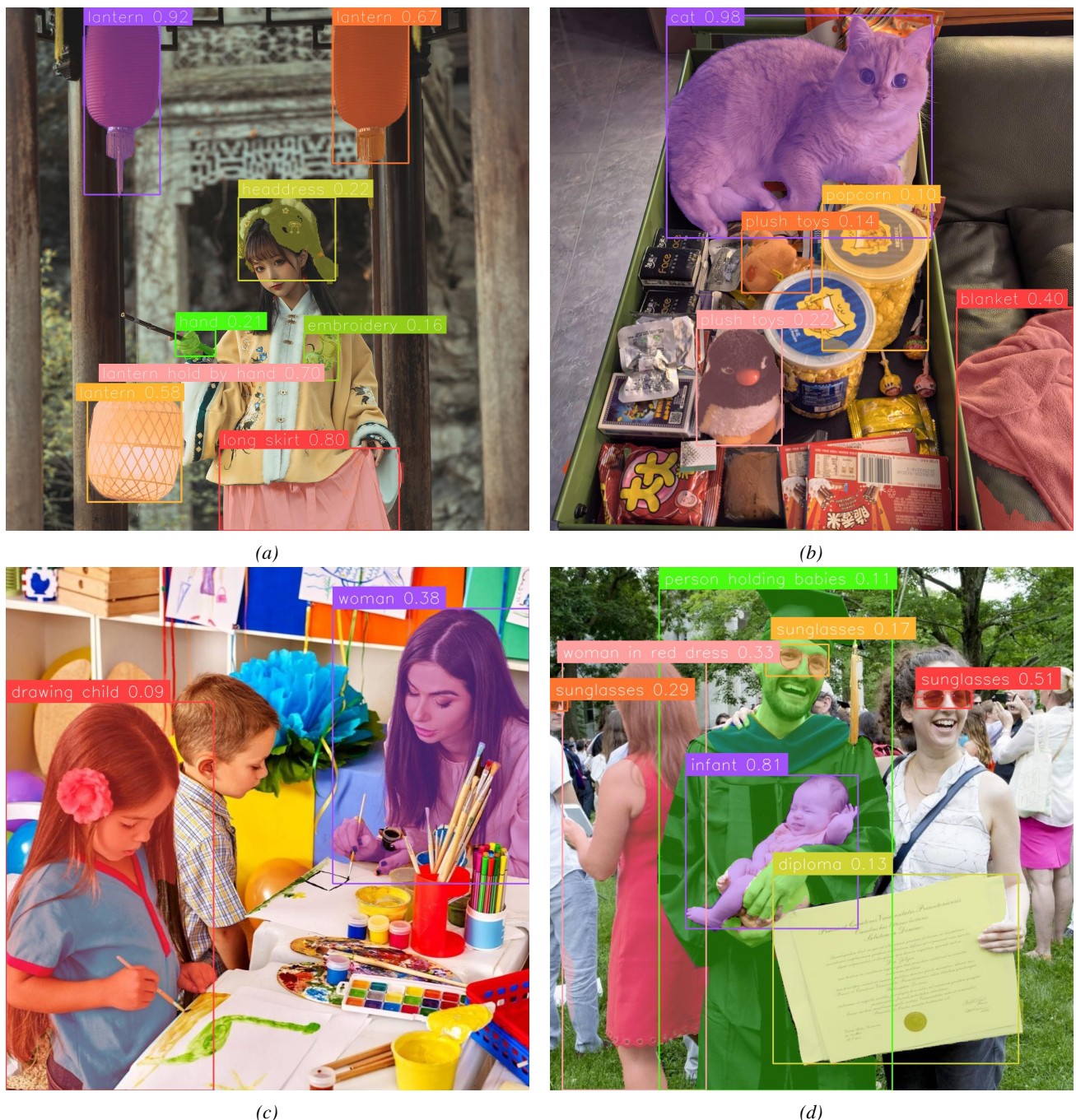

*Figure 6.* (a) prompts: lantern, lantern hold by hand, hand, headdress, embroidery, long skirt. (b) prompts: cat, popcorn, blacket, plush toys. (c) prompts: drawing child, woman. (d) prompts: diploma, person holding babies, woman in red dress, infant, sunglasses. Zoom in for better visual effect.

