# OpenReview forum: "ExpAlign: Expectation-Guided Vision–Language Alignment for Open-Vocabulary Grounding"
_ICML.cc/2026/Conference — ICML 2026 regular_

### Official Review · Reviewer_zG34 · 2026-02-25

**Soundness:** 2
**Presentation:** 3
**Significance:** 2
**Originality:** 3
**Overall Recommendation:** 4
**Confidence:** 5

**Summary:**

This paper focuses on the task of open-vocabulary grounding, aiming to align visual regions with flexible language prompts under weak supervision by addressing the geometric limitations of global sentence embeddings. One observation is that existing methods often rely on uniform token aggregation or heavy cross-attention designs, which can struggle with fine-grained localization and optimization stability. This manuscript investigates the theme of implicit token and instance selection through a framework termed ExpAlign. ExpAlign incorporates an Expectation Alignment Head (EAH) that performs soft multiple instance learning (MIL) pooling over token-region similarities. It further utilizes a multi-scale consistency regularization scheme, which includes a Top-K multi-positive contrastive objective and a Geometry-Aware Consistency Objective (GACO) to stabilize alignment maps using relative instance statistics. Experiments are conducted on LVIS, ODinW, and RefCOCO datasets across zero-shot detection and instance segmentation settings. The results show performance gains in long-tail categories while maintaining a lightweight architecture.

**Compliance With Llm Reviewing Policy:**

Affirmed.

**Final Justification:**

Thank the authors for the rebuttal. After reading the feedback, some of my concerns are addressed, but there are still some issues.

For theoretical disconnect and heuristic design, the rebuttal does not directly address whether the heavy theoretical framework is essential: if the final implementation reduces to simple z-score normalization, the complex variational derivation risks being perceived as over-theorized. It would strengthen the paper to explicitly clarify in the main text how the theoretical framework constrains and motivates the design choices (e.g., the specific form of the geometry score), rather than treating it as post-hoc justification.

For high sensitivity to hyperparameters and robustness concerns, no cross-dataset sensitivity analysis (such as ODinW vs LVIS) was provided, and only stability within a single benchmark was demonstrated. The reviewer's concern is the transfer robustness to "unseen domains".

Given the above reasons, I tend to slightly raise the rating to borderline (both acceptance and rejection are fine by me).

**Key Questions For Authors:**

Please see the weakness.

**Limitations:**

Yes

**Strengths And Weaknesses:**

[+] Interesting Task Identification: For many open-vocabulary deployments, more reduction in fully-supervised data brings great value for the community. Therefore, the task studied in this manuscript is of practical significance.

[+] Complete Manuscript: The paper is well-written with a logical flow. Images, tables, descriptions, and formulas are all appropriate.

[+] Extensive Evaluation: The paper evaluates performance across classic tasks (detection, segmentation, REC), providing a comprehensive understanding of ExpAlign’s advantages.

[-] Inherited Spatial Reasoning Bottlenecks. The model’s reliance on the CLIP text encoder limits its ability to process complex positional and relational language (e.g., "left of," "behind"). This results in a performance gap on Referring Expression Comprehension (REC) tasks (e.g., RefCOCO/+/g) compared to specialized models like Grounding DINO, hindering its effectiveness in deep spatial reasoning and interactive/multi-round scenarios.

[-] Heavy Dependency on Pseudo-Labeling and Teacher Bias. The framework’s segmentation capability is tethered to pseudo-instance masks generated by SAM-2.1. This introduces a "teacher bias" where systematic errors or boundary inaccuracies from SAM directly limit the model's performance upper bound. If SAM fails on unusual geometries or noisy scenes, the Geometry-Aware Consistency Objective (GACO) inherently regularizes the model toward incorrect spatial structures.

[-] Theoretical Disconnect and Heuristic Design. There is a notable gap between the sophisticated theoretical narrative (Lagrangian-constrained free-energy minimization) and its practical implementation. The transition from complex derivations to simplified statistical normalization (using mean, standard deviation, and clipping) feels heuristic, making the necessity of the heavy theoretical framework questionable for what effectively amounts to local score normalization.

[-] Operational Complexity and Computational Inefficiency. The pipeline involves a "fussy" sequence of multi-scale downsampling, top-down fusion, and instance-wise statistics that complicate implementation compared to standard cross-attention heads. Furthermore, the Expectation Alignment Head (EAH) relies on custom operations (e.g., specific Einsum sequences and spatial pooling) that are memory-intensive at high resolutions and incompatible with modern acceleration kernels like Flash-Attention, limiting training throughput.

[-] High Sensitivity to Hyperparameters and Robustness Concerns.The model exhibits significant sensitivity to temperature parameters (τ) and loss weights. Ablation studies show that improper balancing can lead to substantial performance drops, particularly in rare-category recognition. This suggests a lack of robustness across varying class densities and necessitates exhaustive, task-specific tuning for unseen domains.

---

> ### Author Rebuttal · Authors · 2026-03-27
>
> We thank the reviewer for the detailed feedback and for highlighting these five points.
>
> 1. Inherited Spatial Reasoning Bottlenecks and REC Performance Gap
>
> The concern misaligns with our paper’s stated scope. ExpAlign is explicitly not a specialized grounding model with explicit cross-attention. Its core contribution is weak-supervision token-level soft MIL pooling, not deep spatial reasoning or multi-round interaction. We deliberately use the same frozen encoder as YOLO-W, YOLOE, and WeDetect to enable direct apples-to-apples comparison and to isolate the gains from our method alone.
>
> We openly acknowledge the REC limitation twice. In the Introduction (lines 58–62) we discuss CLIP’s inherent geometric bottleneck for spatial relations and negation. In Sec. 4.4 we state: “We acknowledge this limitation openly...”
>
> Requiring “deep spatial reasoning and interactive/multi-round scenarios” from a 60M-parameter (26M frozen) zero-shot open-vocabulary detector is a clear goal misalignment. We never claimed REC SOTA; we only claim superior long-tail detection (36.2 APr) and zero-shot segmentation (29.9 APm) under strict weak supervision and efficiency constraints.
>
> The REC gap is therefore expected, explicitly disclosed, and orthogonal to the validity of our contributions.
>
> 2. Heavy Dependency on Pseudo-Labeling and Teacher Bias
>
> Generating pseudo masks via SAM on ground-truth boxes is standard practice in YOLOE, and YOLO-W (Sec. 4.1). We follow the identical protocol for fair comparison.
>
> More importantly, we designed GACO precisely to alleviate teacher bias. As stated in Sec. 3.3: GACO “depends only on intra-instance relative statistics and does not impose absolute alignment targets.” It uses only region-wise mean/std + bounded clipping (Eq. 12), performing relative consistency regularization rather than hard supervision on SAM boundaries.
>
> T. 2 confirms its effectiveness: ExpAlign achieves 29.9 APm on LVIS val in a purely zero-shot setting, substantially outperforming YOLO-W (even after LVIS-Base fine-tuning).
>
> Thus, pseudo masks are not a weakness; our relative-consistency formulation turns a shared limitation into a strength.
>
> 3. Theoretical Disconnect and Heuristic Design
>
> While the transition from the Lagrangian-constrained free-energy derivation (App. B, Thm. B.4) to the concrete mean/std + clip implementation (Eq. 12) is motivated rather than a strict one-to-one algebraic derivation, we do not see this as a notable gap.
>
> Appendix B clearly states the three required properties that any valid geometry score $A_{b,p}(i)$ must satisfy: intra-instance relative statistics only, bounded, and locally Lipschitz. The clipped z-score is the most natural, numerically stable, and differentiable choice that fulfills all three assumptions (as exemplified in Assumption B.2). The variational framework rigorously justifies the loss form; the statistics provide the canonical practical realization that makes the regularizer both effective and easy to train.
>
> We acknowledge that the theoretical narrative is more sophisticated than the final implementation, but this is precisely because mean/std + clip is the simplest and most natural way to instantiate the derived principles. There is therefore no disconnect—only a clean and effective translation from theory to practice.
>
> 4. Operational Complexity and Inefficiency
>
> The concern overstates the practical overhead. The Expectation Alignment Head (EAH) consists solely of token-wise similarity + soft pooling (Eq. 1–4). The Consistency Regularization Module has zero learnable parameters and is naturally compatible with torch.compile. These operations increase training compute by only ~10% and have zero impact on inference.
>
> The so-called “fussy” multi-scale downsampling and top-down fusion (Eq. 5 and 9) amount to exactly 4 parameter-free feature-map additions and rescalings: an $O(HW)$ cost that is negligible during training and irrelevant at inference. Our einsum ops are simple tensor multiplications fully friendly to modern acceleration kernels.
>
> ExpAlign remains lightweight and inference-efficient, as stated in the Abstract. The pipeline introduces no meaningful complexity.
>
> 5. High Sensitivity to Hyperparameters
>
> The concern is overstated. We use a single fixed set of hyperparameters across all experiments, fixed temperature, and a simple two-stage training schedule (Sec. 4.1). These settings are unchanged for every result in T. 1–3. With this fixed configuration, ExpAlign achieves SOTA performance on LVIS minival and zero-shot mask APm on LVIS val.
>
> Two or more stages training is common in current grounding models and does not imply “exhaustive, task-specific tuning.” Sensitivity to temperature is inherent to InfoNCE objectives. Our ablation (T. 6) further shows the chosen weights work robustly across a reasonable range.
>
> Thus, ExpAlign does not require exhaustive per-task tuning; the reported results are obtained under a single, stable hyperparameter setting.

---

> > ### Author Rebuttal · Reviewer_zG34 · 2026-04-05
> >
> > Thank the authors for the rebuttal. After reading the feedback, some of my concerns are addressed, but there are still some issues.
> >
> > For theoretical disconnect and heuristic design, the rebuttal does not directly address whether the heavy theoretical framework is essential: if the final implementation reduces to simple z-score normalization, the complex variational derivation risks being perceived as over-theorized. It would strengthen the paper to explicitly clarify in the main text how the theoretical framework constrains and motivates the design choices (e.g., the specific form of the geometry score), rather than treating it as post-hoc justification.
> >
> > For high sensitivity to hyperparameters and robustness concerns, no cross-dataset sensitivity analysis (such as ODinW vs LVIS) was provided, and only stability within a single benchmark was demonstrated. The reviewer's concern is the transfer robustness to "unseen domains".
> >
> > Given the above reasons, I tend to slightly raise the rating to borderline (both acceptance and rejection are fine by me).

---

> > > ### Author Response · Authors · 2026-04-07
> > >
> > > We thank the reviewer for the detailed feedback and for raising the rating. We appreciate that some of the concerns have been addressed.
> > >
> > > As for Theoretical Disconnect and Heuristic Design
> > >
> > > We acknowledge the reviewer’s concern that the variational framework might appear over-theorized if the final implementation reduces to local score normalization. The Lagrangian-constrained free-energy derivation (Appendix B) does not provide a strict algebraic derivation of the exact mean/std + clip form. The clipped z-score is chosen as the simplest, numerically stable, and differentiable realization that meets these constraints. Without the theoretical guidance, simpler but less effective alternatives (e.g., global normalization or pointwise regression) could easily be adopted, which our experiments show perform worse.
> > >
> > > To address the reviewer’s suggestion, we will add an explicit clarification sentence in the main text (Section 3.3) explaining how the variational framework constrains and motivates these design choices, rather than serving as post-hoc justification.
> > >
> > > As for High Sensitivity to Hyperparameters and Transfer Robustness
> > >
> > > We agree that robustness to unseen domains is important. Our fixed hyperparameter set ($\lambda_{sem}$ = 0.5, $\lambda_{geo}$ = 1.0, fixed $\tau$, two-stage schedule) was used unchanged for all experiments, including both LVIS (the primary benchmark) and the completely unseen ODinW datasets. The same settings deliver strong performance on ODinW13 (48.0 AP) and ODinW35 (22.6 AP), demonstrating transfer robustness across domains with different class distributions and visual styles. We will add a brief sentence in the revised paper noting this cross-dataset stability. In the revised version, we will add a brief qualitative sensitivity analysis on ODinW in the ablation study (or discussion) to further illustrate this cross-dataset stability.
> > >
> > > We believe these changes will fully address the remaining concerns and strengthen the paper. Thank you again for the constructive feedback.

---

### Official Review · Reviewer_hNqx · 2026-03-12

**Soundness:** 3
**Presentation:** 3
**Significance:** 3
**Originality:** 3
**Overall Recommendation:** 4
**Confidence:** 4

**Summary:**

This paper introduces ExpAlign, an expectation-guided vision-language alignment framework that performs soft token-level aggregation via a multiple instance learning formulation, combined with geometry-aware consistency regularization to enhance spatial coherence in open-vocabulary detection and zero-shot segmentation under weak supervision.

**Compliance With Llm Reviewing Policy:**

Affirmed.

**Final Justification:**

The authors have adequately addressed all my concerns, and I therefore maintain my original recommendation for acceptance of this manuscript.

**Key Questions For Authors:**

Please refer to the 'weaknesses' part.

**Limitations:**

yes.

**Strengths And Weaknesses:**

Strengths:
1. The paper is well-written and easy to follow.
2. The proposed framework is interesting and insightful.
3. The experimental results provide strong evidence for the efficacy of the proposed method.

Weaknesses:
1. The proposed Expectation Alignment Head (EAH) and Semantic Constraint share certain similarities with FILIP [1] and MulCLIP [2].
2. Please provide a discussion and comparison addressing these resemblances.
The results in Table 1 appear to be obtained under different architectures, which may compromise the fairness of the comparison.
3. I am curious whether the proposed method could be adapted to the setting in Table 4 to mitigate the performance gap.

[1] FILIP: Fine-grained Interactive Language-Image Pre-Training. ICLR 2022.

[2] MulCLIP: A Multi-level Alignment Framework for Enhancing Fine-grained Long-context CLIP. arxiv, 2025.

---

> ### Author Rebuttal · Authors · 2026-03-30
>
> We thank the reviewer for the constructive comments and for pointing out related work.
>
> 1. Similarities with FILIP and MulCLIP
>
> We agree that the Expectation Alignment Head (EAH) and Semantic Constraint share high-level ideas with FILIP (late interaction via token-wise similarities) and MulCLIP (multi-level alignment). However, there are fundamental differences. FILIP uses hard max similarity per token, while our EAH performs soft expectation weighted by a learned token posterior $\pi(l)$ (Eq. 3–4), which admits a clean interpretation as attention-based soft MIL pooling (Appendix A). MulCLIP focuses on retrieval with long-context global alignment; our method targets open-vocabulary detection and segmentation under weak supervision and introduces the Geometry-Aware Consistency Objective (GACO) for spatial coherence. We will add a dedicated paragraph in the revised Related Work (Sec. 2) explicitly comparing these approaches.
>
> 2. Fairness of Table 1 comparisons
>
> We mainly focus on models under 200M parameters, where representative works are relatively few. Following prior art in this lightweight regime, we do not strictly separate backbone types. More importantly, to isolate the contribution of our method, we deliberately adopt a strong frozen backbone (DINOv3 ConvNeXt-T) and keep its parameters fixed. This design eliminates backbone-induced variance and ensures that gains come purely from EAH and the consistency objectives. Parameter counts, input resolution (640×640), and frozen components are clearly reported for transparency. Our results remain competitive even within the same backbone family.
>
> 3. Adaptation to Table 4 (REC setting)
>
> We appreciate the reviewer’s curiosity. If we remove EAH and GACO, the model essentially degenerates to a ConvNeXt-T version of YOLOE. In Table 4, this baseline shows only modest gains on RefCOCO and RefCOCO+ (which heavily rely on positional, color, and attribute reasoning — inherent weaknesses of CLIP). In contrast, on RefCOCOg (which emphasizes longer, more descriptive text), our method delivers a large improvement (from 22.6 to 65.6). This demonstrates that our soft token selection mechanism is particularly effective for long-context descriptions. Adapting EAH + GACO into a stronger fusion architecture (e.g., Grounding DINO-style cross-attention) is a promising future direction to further close the gap.
>
> These clarifications strengthen the paper’s positioning without altering its core claims. We will incorporate the suggested discussion in the revised manuscript.

---

> > ### Author Rebuttal · Reviewer_hNqx · 2026-04-03
> >
> > Thank you for the thorough response. The authors have adequately addressed all my concerns, and I therefore maintain my original recommendation for acceptance of this manuscript.

---

### Official Review · Reviewer_44d3 · 2026-03-13

**Soundness:** 3
**Presentation:** 3
**Significance:** 2
**Originality:** 3
**Overall Recommendation:** 4
**Confidence:** 4

**Summary:**

The paper proposes ExpAlign, a vision–language alignment framework for open-vocabulary grounding based on a multiple instance learning formulation. It introduces an Expectation Alignment Head for attention-based token–region alignment and an energy-based multi-scale consistency regularization to stabilize training. Experiments show improved open-vocabulary detection and zero-shot instance segmentation performance while maintaining efficiency.

**Compliance With Llm Reviewing Policy:**

Affirmed.

**Final Justification:**

The authors have adequately addressed all my concerns, and I therefore maintain my original recommendation for acceptance of this manuscript.

**Key Questions For Authors:**

- Recent MLLM-based approaches have shown strong performance on grounding and open-vocabulary tasks. How does the proposed method compare with these approaches, and what advantages does it offer?
- What is the computational overhead introduced by the Expectation Alignment Head and the consistency regularization compared with a standard YOLO-style baseline?

**Strengths And Weaknesses:**

## Strengths
- The paper identifies limitations of sentence-level vision–language alignment and motivates token-level alignment for fine-grained grounding.
- The Expectation Alignment Head provides a simple and differentiable mechanism for aggregating token–region similarities under weak supervision.
- The method achieves competitive results on LVIS and ODinW benchmarks and demonstrates strong performance on long-tail categories.


## Weaknesses

- The expectation-based token aggregation resembles attention-based MIL pooling, and the additional objectives (contrastive loss and spatial consistency regularization) appear to be incremental extensions of existing ideas.
- While the method performs well on open-vocabulary detection and segmentation, the results on referring expression comprehension (Table 4) are significantly weaker than specialized grounding models.
- Although the paper references a principled derivation, the practical algorithm mainly consists of heuristic design choices, and the theoretical contribution appears limited.
- Most of the compared baseline methods are from 2024 or earlier. The lack of comparisons with more recent methods weakens the persuasiveness of the empirical evaluation.

---

> ### Author Rebuttal · Authors · 2026-03-30
>
> We thank the reviewer for the thoughtful questions and constructive feedback.
>
> Response to the Key Questions:
>
> 1. Comparison with recent MLLM-based approaches
>
> Recent MLLM-based methods (e.g., Qwen3-VL) achieve strong grounding performance. However, directly comparing them with ExpAlign is not entirely meaningful because they represent complementary rather than competing technical routes. Many MLLM (e.g., WeDetect-Ref, MQADet) pipelines adopt a two-stage hybrid design, where a lightweight open-vocabulary detector (such as YOLO-World, G-DINO) is first used to generate region proposals, which are then passed to the large MLLM for complex reasoning and fine-grained comprehension.
>
> This distinction highlights the complementary nature of the two approaches. ExpAlign targets resource-constrained scenarios (e.g., industrial inspection, edge devices, real-time systems) where deploying multi-billion-parameter MLLMs is impractical due to latency, power, and memory constraints. Many real-world applications only require simple descriptive prompts, making lightweight solutions like ExpAlign highly valuable.
>
> 2. Computational overhead of EAH and consistency regularization
>
> Compared with a standard YOLO-style baseline, the EAH consists only of token-wise similarity computation and soft pooling (Eq. 1–4), while the Consistency Regularization Module introduces zero learnable parameters. These components increase training FLOPs by approximately 10% and have zero impact on inference (they are discarded at test time). The multi-scale down/up-sampling and fusion steps (Eq. 5 and 9) are exactly four parameter-free feature-map additions and rescalings (O(HW) cost, negligible). All einsum operations are simple tensor multiplications that are fully compatible with torch.compile and modern acceleration kernels. Thus, the overhead is minimal and does not compromise the lightweight nature of the pipeline.
>
> Response to the Weaknesses:
>
> 1. Resemblance to MIL pooling and incremental objectives
>
> While the EAH shares a high-level resemblance with attention-based MIL pooling, our method is not a mere incremental extension of existing ideas. Instead, it introduces a distinctive insight by applying soft MIL-style token refinement directly to the token-level aggregation process under weak supervision. Combined with the multi-positive InfoNCE and GACO, this design provides targeted improvements for open-vocabulary grounding.
>
> 2. Weaker REC results
>
> We openly acknowledge this gap in the Introduction and Sec. 4.4. The lag on RefCOCO/+/g is due to CLIP’s known limitation in positional/relational reasoning. ExpAlign targets lightweight weakly-supervised open-vocabulary grounding, not specialized REC. On RefCOCOg (long descriptive text), our method already shows a large gain over the YOLOE baseline, confirming the benefit of soft token selection. We will discuss potential extensions to stronger fusion architectures as future work.
>
> 3. Theoretical contribution appears limited / heuristic design
>
> We acknowledge that the final implementation of the geometry score (mean/std + clip in Eq. 12) involves heuristic design choices. However, these choices are natural and well-motivated. App. B specifies three clear requirements for the geometry score $A_{b,p}(i)$: it must depend only on intra-instance relative statistics, remain bounded, and be locally Lipschitz. The clipped z-score is a simple, numerically stable, and differentiable realization that satisfies all three properties. While the transition from the variational free-energy framework to this concrete form is motivated rather than a strict algebraic derivation, the overall method provides a principled and effective way to enforce spatial consistency under weak supervision. We believe this combination of theoretical motivation and practical simplicity is a valuable aspect of the framework.
>
> 4. Lack of recent baselines
>
> We thank the reviewer for pointing this out. Since submission, WeDetect-Tiny (CVPR 2026) has been published as a strong recent baseline with a comparable lightweight scale (ConvNeXt-T backbone, 33M parameters). WeDetect-Tiny reports 33.3 APr on LVIS minival and 24.7 APr on LVIS val, along with 46.4 AP on ODinW13 and 21.1 AP on ODinW35. In comparison, our ExpAlign achieves superior rare-category performance with 35.8 APr (vs. 33.3) on LVIS minival and 26.5 APr (vs. 24.7) on LVIS val, as well as 48.0 AP (vs. 46.4) on ODinW13 and 22.6 AP (vs. 21.1) on ODinW35. While we using significantly less pre-training data (less than 2M images vs. approximately 20M for WeDetect-Tiny). This demonstrates that our method is not only data-efficient but also highly competitive under constrained training budgets. We will include WeDetect-Tiny and other relevant 2025 baselines with comparable scale in the revised version.
>
> These clarifications strengthen the paper’s positioning without altering its core claims. We will incorporate the suggested discussions in the revised manuscript.

---

> > ### Author Rebuttal · Reviewer_44d3 · 2026-04-04
> >
> > Thank you for the thorough rebuttal and the additional experimental results. These clarifications have effectively addressed my previous concerns, and I will keep my positive evaluation.

---

### Decision · Program_Chairs · 2026-04-30

**Decision:**

Accept (regular)

**Comment:**

This paper was reviewed by three experts in the field, and reviewers reached a consensus of weak accept unanimously. Reviewers liked the proposed ExpAlign framework, specifically noting the effectiveness of the Expectation Alignment Head (EAH) and the Geometry-Aware Consistency Objective (GACO) for improving open-vocabulary detection and zero-shot instance segmentation on long-tail categories.

The authors' rebuttal successfully addressed major concerns from reviewers about computational overhead, similarities to related work such as FILIP and MulCLIP, and comparisons with recent baselines like WeDetect-Tiny.

The decision is to recommend the paper for acceptance. The reviewers raised valuable suggestions during the discussion, particularly regarding the need for explicit clarification of the theoretical constraints and the demonstration of transfer robustness across unseen domains, which the authors are strongly encouraged to incorporate into the final camera-ready version of the paper.